# Mixture-of-Instructions: Aligning Large Language Models via Mixture Prompting

## Abstract

With the proliferation of large language models (LLMs), the comprehensive alignment of such models across multiple tasks has emerged as a critical area of research. Existing alignment methodologies primarily address single task, such as multi-turn dialogue, coding, mathematical problem-solving, and tool usage. However, AI-driven products that leverage language models usually necessitate a fusion of these abilities to function effectively in real-world scenarios. Moreover, the considerable computational resources required for proper alignment of LLMs underscore the need for a more robust, efficient, and encompassing approach to multi-task alignment, ensuring improved generative performance. In response to these challenges, we introduce a novel technique termed Mixture-of-Instructions (MoI), which employs a strategy of instruction packing combined with diverse system prompts to boost the alignment efficiency of language models. We have also compiled a diverse set of seven benchmark datasets to rigorously evaluate the alignment efficacy of the MoI-enhanced language model. Our methodology was applied to the open-source Qwen-7B-chat model, culminating in the development of Qwen-SFT-MoI. This enhanced model demonstrates significant advancements in generative capabilities across coding, mathematics, and tool use tasks.

## 1 Introduction

The rise of large language models (LLMs) has underscored the importance of effective training techniques, encompassing Pre-training, Supervised Fine-Tuning (SFT), and Reinforcement Learning from Human Feedback (RLHF), to expand their knowledge and ensure alignment with human values for greater reliability. However, despite advancements, LLMs often falter in specialized tasks such as coding, mathematics, and reasoning. With the growing access to high-quality, domain-specific datasets(Luo et al., 2023c; Li et al., 2023a; CodeFuse, 2023; Wei et al., 2023; Yue et al., 2023; Yu et al., 2023; Luo et al., 2023a; Zeng et al., 2023; Qin et al., 2023), this study investigates how targeted SFT can boost LLM capabilities in these areas while preserving existing competencies.

We examined a case where the Qwen-7B-chat (Bai et al., 2023)model failed to learn Python code for the Boyer-Moore majority vote algorithm. Training with standard code dataset proved ineffective, with the model producing suboptimal solutions due to prompt-induced knowledge conflicts. To our surprise, we found that simply changing to a different system prompt and retraining could easily overcome this issue, indicating that altering prompts can resolve knowledge conflicts and improve task performance across tasks.

Leveraging our initial findings, we developed a strategy that assigns unique system prompts to different tasks and integrates these varying prompts into a unified instruction set for model alignment. We introduce the Mixture of Instructions (MoI) methodology, a structured approach to amalgamate open-source datasets for comprehensive multi-task training, spanning dialogue, code, math, and agent-based tasks, each with bespoke prompts. This not only recasts training as a multi-task learning (Chung et al., 2024; Wei et al., 2021; Raffel et al., 2020; Sanh et al., 2021) endeavor but also tackles challenges such as dataset bias, where improvement in one task could detriment another. Through MoI, we adeptly use diverse prompts to ensure balanced task performance, effectively navigating the complexities of multi-task learning.

Ultimately, we integrated the MoI with multi-turn dialogue, code generation, mathematics reasoning and tool usage datasets for comprehensive training, resulting in a significantly enhanced language

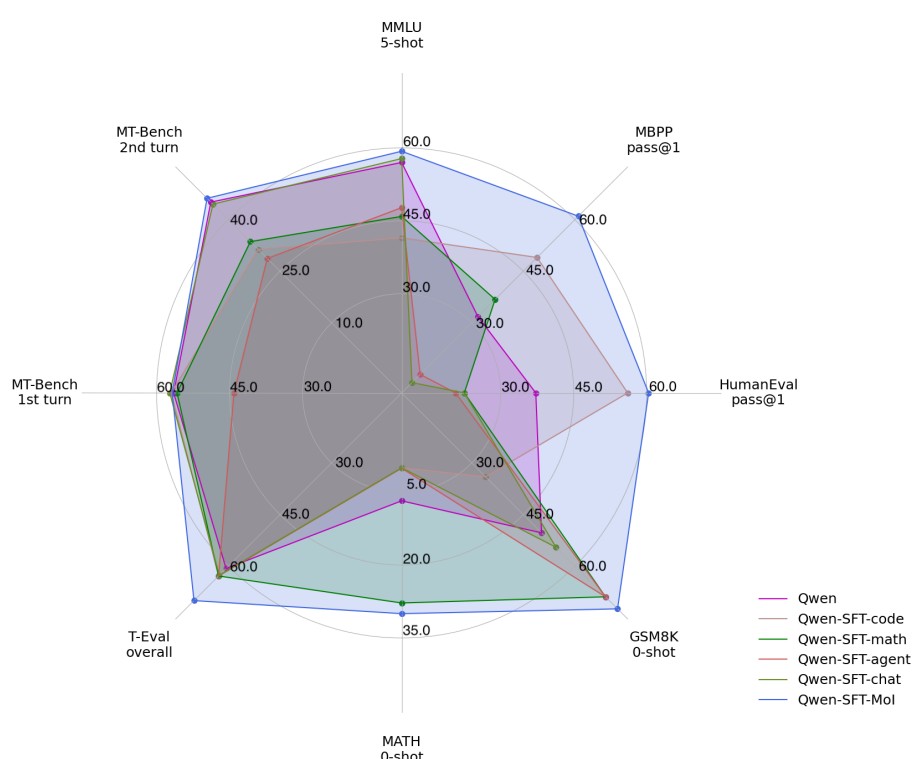

Figure 1: Performance of Qwen-SFT-MoI and Qwen-7B-chat models, along with various SFT-aligned models on subdomain datasets, evaluated across seven datasets encompassing mathematics, programming, tool usage, common sense, and both single and multi-turn dialogues. Results demonstrate that training with our MoI method enhances multiple capabilities of language models, achieving improved alignment.

model—Qwen-SFT-MoI. This model exhibited substantial improvement across a range of benchmarks, including MT-Bench(Zheng et al., 2024a), HumanEval(Chen et al., 2021), MBPP(Austin et al., 2021), MATH(Hendrycks et al., 2021), GSM8K(Cobbe et al., 2021), MMLU(Hendrycks et al., 2020), and T-EVAL(Chen et al., 2023), compared to the Qwen-7B-Chat model, validating our MoI strategy for further alignment on an already SFT model. To summarize, our work makes the following contributions:

- We identified and addressed conflicts between new and old knowledge during SFT, by introducing modified system prompts that aid in the integration of new knowledge. This resulted in improved alignment with and performance on code generation task.

- Our MoI method facilitates joint multi-task training, effectively reducing dataset bias and preserving conversational abilities. The efficacy of the MoI approach was substantiated through extensive experimentation.

- By applying MoI alongside high-quality datasets on the Qwen-7B-chat model, we developed the Qwen-SFT-MoI model. This model demonstrates superior performance in mathematical reasoning, code generation, tool usage, and chat benchmarks, thereby highlighting the effectiveness of MoI in enhancing SFT models.

## 2 MIXTURE OF INSTRUCTIONS

### 2.1 SYSTEM PROMPT MATTERS

Our motivation began with testing the performance of the Qwen-7B-chat(Bai et al., 2023) model on the MT-Bench(Zheng et al., 2024a) dataset. A substantial amount of study(Hu et al., 2024; Zhang et al., 2024; Inc., 2023; Chiang et al., 2023) indicates that small language model do not perform well

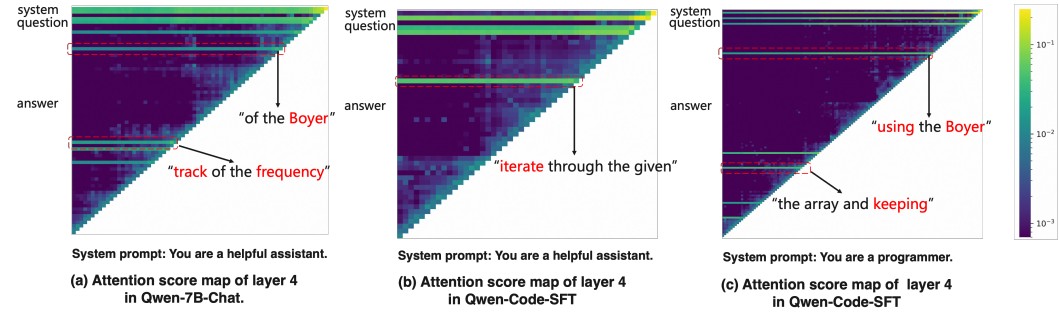

Figure 2: Attention maps for responses to Question ID 127 in the MT-Bench, which show attention distribution across the system prompt, the question, and each model's answer. (a) Qwen-7B-chat focuses heavily on the prompt and question but incorrectly associates 'Boyer' with frequency tracking, misrepresenting the Boyer-Moore algorithm. (b) After SFT on code generation data, the model still overemphasizes the prompt and question, overlooks 'Boyer' and fixates on iterative element search. (c) Post-SFT with a new system prompt, the model shifts attention towards generating an answer, correctly hinting at the Boyer-Moore algorithm, which fundamentally tracks a candidate element.

on mathematical and reasoning tasks within the MT-Bench. The Qwen-7B-chat also did not achieve high scores on the coding questions in the MT-Bench, particularly to specific question: *Write a function to find the majority element in a given integer array using the Boyer-Moore Voting Algorithm.* Although Qwen-7B-chat successfully identified the majority element, its approach resulted in a higher time complexity of O(nlog(n)) and a space complexity of O(n), deviating from the expected O(n) time and O(1) space complexities of the algorithm. Subsequently, we discovered this question corresponds to an original problem from the LeetCode repository. Consequently, we downloaded 2612 LeetCode problems along with their solutions and trained the Qwen-7B-chat model using SFT. However, we observed that the model, even after SFT, still failed to answer this particular question.

In the current mainstream SFT schemes, an instruction is defined as consisting of three parts: system, question, and answer. As depicted in Figure 2, analysis of attention maps for both the original Qwen-7B-chat and SFT-enhanced models when processing a specific question revealed that the original Qwen-7B-chat model favored array traversal and frequency counting, divergent from the Boyer-Moore algorithm's approach. The SFT model also disproportionately focused on array operations and unduly on the system prompt.

To redirect the attention of a large language model, we replaced its system prompt with *'You are a programmer'* in the SFT regime, inspired by Wang et al. (2024). This change, as shown in Figure 2, led to a realigned attention map that concentrated on the key 'Boyer-Moore Algorithm' principles, prompting accurate descriptions of the algorithm's iteration and candidate tracking. The shift in focus from the prompt to the algorithmic core problem and solution was significant. Additionally, we evaluated the effects of varying SFT-driven system prompts on model performance using the HumanEval, MBPP, and MT-Bench coding benchmark. Table 3 conveys that the *'You are a programmer'* prompt engendered a notable uplift in code generation proficiency.

## 2.2 PACKING INSTRUCTIONS WITH BALANCED SAMPLING

Building on the effectiveness of system prompts, we investigate whether dividing the training dataset into multiple tasks and setting specific system prompts for each improves model alignment. Multi-task learning, as shown by numerous studies (Radford et al., 2019; Sanh et al., 2021; Chung et al., 2024), demonstrates that pre-trained + fine-tuned language models inherently possess multi-task capabilities. However, when aggregating datasets from multiple tasks for sequence training, we found that models trained on combined tasks performed worse than those trained on individual tasks as shown in Table 2. This highlights a common issue in multi-task learning: training on combined data can lead to model bias towards certain tasks, enhancing performance in some while degrading it in others.

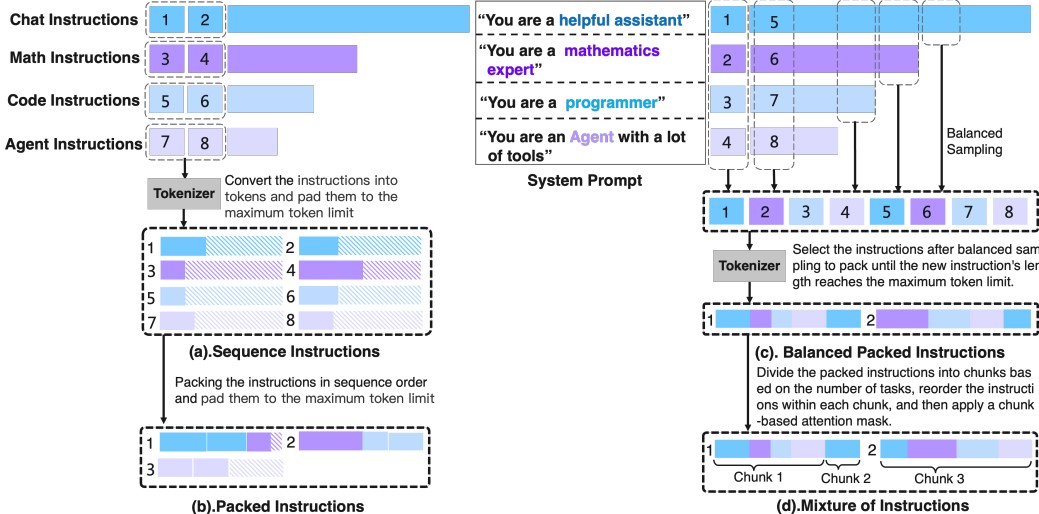

Figure 3: (a). Sequence of Instructions involves extracting instructions from a dataset, tokenizing them, and using padding tokens to reach a fixed maximum length for SFT in an LLM. (b). Packed Instructions merges several instructions into a longer single instruction, minimizing the need for padding and enhancing training efficiency. (c).Balanced Packed Instructions employs balanced sampling from various datasets and concatenates instructions to meet a maximum token length. (d). Mixture of Instructions then prioritizes the instruction with the default system prompt at the start.

A direct approach to address this issue is to increase the batch size during model training(McCandlish et al., 2018), thereby obtaining more balanced gradients, which optimizes the model's alignment. In addition to directly increasing the batch size, packing multiple data samples into a single one (Wei et al., 2021; Raffel et al., 2020; Sanh et al., 2021; Chung et al., 2024) can implicitly achieve the effect of a larger batch size. After applying the packing technique, we found that the model performed better on both mathematical and coding tasks. However, its performance on the MT-Bench declined, indicating that there is still room for improvement in the standard data packing approach.

To address this, we proposed a balanced sampling scheme on packed data. By using a resampling method (Ming et al., 2018), we ensure that the amount of data from different tasks within a single packed instruction is as balanced as possible. The structure of the balanced sampling method is illustrated in the Figure 3 which sequentially collects instructions from datasets of various tasks, decodes these instructions into tokens, and combines them. The effects of using balanced sampling are shown in Table 2.

## 2.3 CHUNK-BASED ATTENTION MASKING

Although a balanced sampling scheme can mitigate dataset biases and promote domain-specific learning in models, both packing and balanced sampling methods combine multiple training examples into a single sequence without applying an attention mask to prevent tokens from attending across packed example boundaries. This phenomenon, known as attention cross-contamination(Krell et al., 2021; Zhao et al., 2024), can affect model performance. Our experiments, shown in Table 8, indicate that attention cross-contamination lowers the model's performance on complex reasoning tasks. However, it can also enhance the model's abilities in code and mathematical tasks.

Zhao et al. (2024) reveals that achieving optimal training results during pre-training requires making the packed training examples as similar as possible and isolating them using attention masks. Our proposed chunk-based attention masking approach shares similarities with this scheme, but we construct chunks within each balanced sampled packed instruction based on the number of domains (such as the four domains in this paper) and ensure that each chunk is sorted in the same order. By applying chunk-based attention masking, we block interference between different chunks while ensuring high similarity between the data in each pair of chunks. Our experimental results, shown in

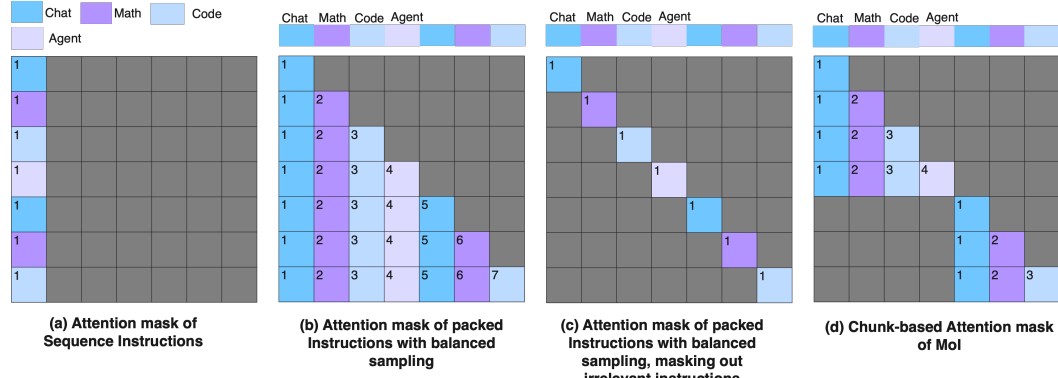

Figure 4: The **numbers** represent the simplified position IDs, which are used to generate the masked-out position embedding. Comparison of different attention masks on the same data: (a) default attention mask for sequence instruction concatenation, (b) default attention mask for balanced sampling concatenation, (c) specially designed mutually isolated attention mask, and (d) our chunk-based attention mask.

Table 8, demonstrate that by using chunk-based attention, the mathematical and coding capabilities derived from packing are preserved while also enhancing the model's complex reasoning ability. Figure 4 provides a schematic representation of the chunk-based attention mask implementation.

## 2.4 MIXTURE OF INSTRUCTIONS

Ultimately, we propose the Mixture of Instructions(MoI) scheme, which comprises several key components. First, we set unique system prompts for each domain-specific task to ensure tailored guidance. Next, we employ balanced sampling techniques to concatenate task data from multiple domains into proportionally balanced datasets. During training, we apply a chunk-based attention mask to isolate unrelated contexts, thereby achieving optimal alignment. By explicitly modifying the optimization objective of standard SFT, MoI achieves superior multi-task learning alignment.

During the standard SFT processing, language models can learn alignment by fine-tuning on high-quality pairs of system prompt $s$, question $x$ and answer $y$. Processed through an auto-regressive LLM weights $\theta$ to maximize the probability $P_\theta(y|[s,x])$. The loss function of SFT should be:

$$\mathcal{L}_{\text{SFT}}(\theta) = -\mathbb{E}[\log P_\theta(y \mid [s,x])] \tag{1}$$

Specifically, in the implementation process, we compute the loss for each data point and then divide it by the number of tokens in the response. Then the loss of each batch in following format:

$$\mathcal{L}_{seq([s,x,y])}(\theta) = -\frac{1}{\sum_{n=1}^{N}|y_n|}\sum_{n=1}^{N}\log P_\theta(y_n \mid [s,x]_n) \tag{2}$$

where $N$ indicates the batch size number, $|y|$ denotes the number of tokens in the response and the $[\cdot]$ operation represents token concatenation. In computing the loss function for packed data, we followed a loss calculation scheme that avoids the suppression of the loss for shorter responses in longer packed sequences.

$$\mathcal{L}_{packed([s,x,y])}(\theta) = -\frac{1}{\hat{N}}\sum_{n=1}^{\hat{N}}\sum_{i=L_n}^{L_{n+1}}\frac{\log P_\theta(y_i \mid [[s,x,y]_{<i}, [s,x]_i])}{(L_{n+1}-L_n)\cdot|y_i|} \tag{3}$$

Here, $L_n$ represents the index number of the packed data within the original dataset, $\hat{N}$ represents the batch size of the packed data. Despite calculating the probability of obtaining $y$ based on $s$ and $x$, the model implicitly treats all preceding packed data $[s,x,y]$ as part of the context, thereby interfering with the model's alignment. We additionally generate an attention mask and position ID for each instruction during packing, thereby eliminating the impact of $[s,x,y]$ on the loss.

$$\mathcal{L}_{maskingout([s,x,y])}(\theta) = -\frac{1}{\hat{N}}\sum_{n=1}^{\hat{N}}\sum_{i=L_n}^{L_{n+1}}\frac{\log P_\theta(y_i \mid [s,x]_i)}{(L_{n+1}-L_n)\cdot|y_i|} \tag{4}$$

Specifically, in an ideal scenario where the number of instructions in each new packed instruction and the token length of each instruction's response are constant, The form of the loss function will become identical to that in sequence training 2:

$$\mathcal{L}_{maskingout([s,x,y])}(\theta) = -\frac{1}{\sum_{i=1}^{N} |y_i|} \sum_{i=1}^{N} logP_\theta(y_i \mid [s,x]_i) \tag{5}$$

By combining $L_{packed}$ and $L_{maskingout}$, the loss function of the MoI scheme by employing a chunk-based attention mask:

$$\mathcal{L}_{MoI([s,x,y])}(\theta) = -\frac{1}{\hat{N}} \sum_{n=1}^{\hat{N}} \sum_{i=\frac{L_n}{n_{mix}}}^{\frac{L_{n+1}}{n_{mix}}} \sum_{j=i\cdot n_{mix}}^{(i+1)\cdot n_{mix}} \frac{logP_\theta(y_j \mid [[s,x,y]_{i\cdot n_{mix}<j}, [s,x]_j])}{(L_{n+1} - L_n) \cdot |y_j|} \tag{6}$$

$n_{\mathrm{mix}}$ is defined as the number of instructions in a chunk. Consequently, when calculating the loss, the model only considers the context within the same chunk, minimizing cross-contamination of attention. MoI refines a single packed data sequence into smaller chunks, creating a multi-level mixture of packed data. This methodology is the basis for the term 'Mixture of Instructions'.

## 3 EXPERIMENTS

### 3.1 EXPERIMENT SETUP

All of our experiments are performed on 8x80G A100 GPUs, using the Qwen-7B-chat model as the base for each independent experiment.We utilized the Huggingface Transformers(Wolf et al., 2019) library to execute our training. During training, we employed DeepSpeed(Rasley et al., 2020) Zero3 optimization and Flashattention-2 (Dao, 2023) to enhance memory efficiency.

### 3.2 TRAINING DATASET

we gathered open-source datasets for four tasks: multi-turn dialogue, code generation (Ding et al., 2023; Wei et al., 2023; Zheng et al., 2024b; Luo et al., 2023c) mathematics reasoning (Yue et al., 2023; Yu et al., 2023), and tool usage (Yang et al., 2024; Qin et al., 2023; Zeng et al., 2023), providing a specific system prompt for each, The detailed information of the data is presented in Table 1.

Table 1: All of our training data are sourced from open datasets downloaded from Huggingface, categorized into four tasks: multi-turn dialogue, code generation, mathematics reasoning, and tools usage. Custom system prompts were individually crafted for the dataset of each task to ensure precise alignment with their respective content.

| Dataset | Task | System Prompt |
|---|---|---|
| Ultrachat200k | Multi-turn dialogue | You are a helpful assistant. |
| MagicCoder110k, EvolInstructCode80k, CodeFeedback157k | Code generation | You are a programmer. |
| MathInstruct262k, MetaMath395k | Mathematics reasoning | You are a mathematics expert. |
| GPT4Tools75k, ToolbenchG3, AgentInstruct | Tool usage | You are an Agent with a lot of tools. |

### 3.3 EVALUATION BENCHMARK

The validation datasets used in this paper include: GSM8K (Cobbe et al., 2021),MATH dataset (Hendrycks et al., 2021),SVAMP (Patel et al., 2021), ASDiv (Miao et al., 2021),HumanEval (Hendrycks et al., 2021) MBPP (Austin et al., 2021),T-EVAL (Chen et al., 2023),MMLU (Hendrycks et al., 2020),BBH-hard (Suzgun et al., 2022) and MT-Bench (Zheng et al., 2024a). For detailed description, please refer to A.1.

### 3.4 ABLATION STUDY AND ANALYSIS

**The impact of different system prompts.** By modifying system prompts for SFT, we can comprehensively enhance the alignment performance of models. As shown in Table 3, the experimental

Table 2: Performance of SFT models on widely-used benchmarks. We report zero-shot performance of the models. For all benchmark computations, we consistently used the system prompt **"You are a helpful assistant."** to ensure the effectiveness of our methodology. When computing the average score, we multiplied the mt-bench scores by 10 to ensure numerical validity. The comparative values in the data are all benchmarked against Qwen-7B-chat. For detailed description, please refer to A.2.

| Model | MMLU 0-shot | MBPP pass@1 | HumanEval pass@1 | GSM8K 0-shot | MATH 0-shot | T-Eval overall | MT-Bench 1st turn | MT-Bench 2nd turn | Avg |
|---|---|---|---|---|---|---|---|---|---|
| Qwen-7B-chat | 57.0 | 31.8 | 37.2 | 50.3 | 6.8 | 60.7 | 5.65 | 4.51 | 43.1 |
| Qwen-SFT-code | 41.5(−15.5) | 53.0(+21.2) | 56.1(+18.9) | 34.0(−16.3) | 0.0(−6.8) | 22.3(−38.4) | 5.70(+0.05) | 3.13(−2.38) | 36.9(−6.2) |
| Qwen-SFT-math | 45.9(−11.1) | 36.8(+5.0) | 22.5(−14.7) | 71.8(+21.6) | 27.8(+21.0) | 14.0(−46.7) | 5.58(−0.07) | 3.36(−1.15) | 38.5(−4.6) |
| Qwen-SFT-agent | 47.7(−9.3) | 15.0(−16.8) | 20.7(−16.5) | 16.1(−34.2) | 0.1(−6.7) | 47.1(−13.6) | 4.41(−1.24) | 2.87(−1.64) | 27.4(−15.7) |
| Qwen-SFT-chat | 57.8(+0.8) | 12.6(−19.2) | 22.5(−14.7) | 54.5(+4.2) | 0.0(−6.8) | 60.8(+0.1) | 5.73(+0.07) | 4.45(−0.06) | 38.7(−4.4) |
| Qwen-SFT-seq | 59.2(+2.2) | 46.6(+14.8) | 50.6(+13.4) | 69.2(+18.9) | 30.3(+23.5) | 69.6(+8.9) | 5.81(+0.16) | 4.60(+0.09) | 53.7(+10.6) |
| Qwen-SFT-packed | 59.7(+2.7) | 49.0(+17.2) | 56.1(+18.9) | 71.3(+21.0) | 30.3(+23.5) | 67.1(+6.4) | 5.99(+0.34) | 4.20(+0.05) | 54.4(+11.3) |
| Qwen-SFT-balanced | 59.4(+2.4) | 51.6(+19.8) | 55.5(+18.3) | 71.9(+21.6) | 30.0(+23.2) | 66.9(+6.2) | 6.12(+0.47) | 5.01(+0.50) | 55.8(+12.7) |
| Qwen-SFT-MoI | 59.3(+2.3) | 61.0(+29.2) | 60.4(+23.2) | 74.5(+24.5) | 31.9(+25.1) | 68.2(+7.5) | 6.91(+1.26) | 4.75(+0.24) | 59.9(+14.2) |

Table 3: Performance of SFT models with different system prompt on widely-used code generation and mathematics benchmarks. We report both zero-shot and few-shot performance of the models.

| Model | System prompt | MBPP Pass@1 | HumanEval Pass@1 | MT-Bench Coding 1st turn | GSM8K Accuracy(%) | MATH Accuracy(%) | MT-Bench Math 1st turn |
|---|---|---|---|---|---|---|---|
| Qwen-7B-chat | Assistant | 31.8 | 37.2 | 3.8 | 50.3 | 6.8 | 2.65 |
| Qwen-SFT-code | Assistant | 53.0 | 56.1 | 4.1 | 34.0 | 0.0 | 2.3 |
| Qwen-SFT-code | Programmer | 55.2 | 60.4 | **5.0** | 32.8 | 0.0 | 2.1 |
| Qwen-SFT-math | Assistant | 36.8 | 22.5 | 1.6 | 71.8 | 27.8 | 4.9 |
| Qwen-SFT-math | Mathematics expert | 34.9 | 20.3 | 1.7 | 72.1 | 30.0 | **5.0** |
| Qwen-SFT-MoI | Assistant | **61.0** | **60.4** | 4.7 | **74.5** | **31.9** | 5.0 |

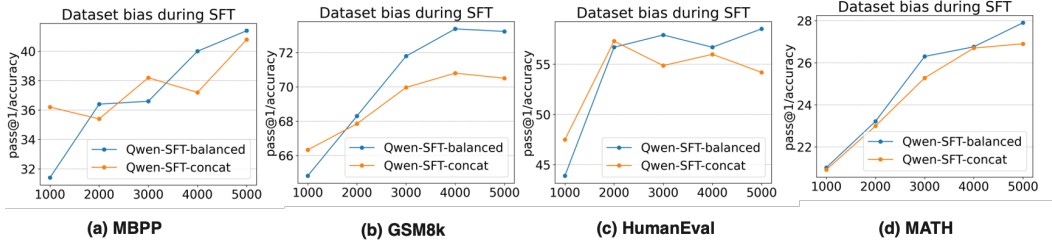

Figure 5: Qwen-SFT-balanced represents the results of data concatenation using balanced sampling, while Qwen-SFT-concat represents the results of regular packed data concatenation. (a) Performance on MBPP dataset using two methods across iteration steps. (b) Comparative performance on GSM8K dataset with two approaches over iterations. (c) Performance evaluation on HumanEval dataset using two methods across iterations. (d) Performance trajectory on MATH dataset under two methodologies over iteration steps.

results indicate that MoI can transfer abilities originally elicited only under their respective system prompts to the default system prompt.

**The impact of training on different task datasets.** Our comparative ablation study in Table 2 revealed that while models trained on math, and code data respectively enhance performance in their specific tasks, this specialization comes at the cost of diminished multi-turn conversation ability, as evidenced by a significant drop in MT-Bench 2nd turn scores for all models. Thus, models trained on these datasets are recommended for domain-specific tasks rather than as conversational assistants. Additionally, models trained on the agent dataset did not show improvement in any metric, suggesting that the T-Eval dataset serves as a comprehensive test scheme to assess model capabilities.

**Dataset bias.** Figure 5 shows our ablation study compared Qwen-SFT-packed and Qwen-SFT-balanced, demonstrating that balanced sampling rapidly improves GSM8K accuracy and stabilizes code generation performance during training. In contrast, packing showed no substantial late-stage gains in GSM8K and a decline in code capability. With identical datasets and hyperparameters, these differences highlight balanced sampling superior handling of multi-task learning in LLM.

**The impact of different data sampling strategies.** Based on Table 2, it can be observed that the sequence strategy significantly enhances the model's performance across all validation datasets. The packing strategy, building upon the sequence sampling approach, further improves the model's performance on mathematical and coding tasks, yet it diminishes the model's ability in conducting

Table 4: Scores on HumanEval, MBPP, GSM8K, MATH. The pass@1 scores of our models are computed with greedy decoding. Models are evaluated in 0-shot on all datasets.

| Model | Params | HumanEval pass@1 | MBPP pass@1 | GSM8K Accuracy(%) | MATH Accuracy(%) |
|---|---|---|---|---|---|
| GPT-3.5(OpenAI, 2022) | - | 73.2 | 81.7 | 80.8 | 35.5 |
| GPT-4(OpenAI, 2023) | - | 86.6 | 83.0 | 92.0 | 42.5 |
| StarCoder Base(Li et al., 2023a) | 7B | 24.4 | 33.1 | - | - |
| Code Llama(Rozière et al., 2023) | 7B | 33.5 | 41.4 | - | - |
| Code Llama-Instruct(Rozière et al., 2023) | 7B | 34.7 | 44.4 | - | - |
| Code Llama-Python(Rozière et al., 2023) | 7B | 38.4 | 47.6 | - | - |
| WizardCoder(Luo et al., 2023c) | 7B | 48.2 | 56.6 | - | - |
| Magicoder(Wei et al., 2023) | 7B | 60.4 | **64.2** | - | - |
| WizardMath(Luo et al., 2023a) | 7B | 50.0 | 53.9 | 54.9 | 10.7 |
| MetaMath(Yu et al., 2023) | 7B | - | - | 66.4 | 19.4 |
| MAmmoTH(Yue et al., 2023) | 7B | 24.4 | 42.4 | 59.4 | 33.4 |
| Llama2(Touvron et al., 2023b) | 7B | 12.2 | 20.8 | 16.7 | 3.3 |
| Mistral(Jiang et al., 2023) | 7B | 27.4 | 38.6 | 47.5 | 11.3 |
| EURUS-SFT(Yuan et al., 2024) | 7B | 55.5 | 59.1 | - | 32.6 |
| Qwen(Bai et al., 2023) | 7B | 37.2 | 35.8 | 52.5 | 10.3 |
| Qwen-SFT-MoI (ours) | 7B | **60.4** | 61.0 | **74.5** | 31.9 |

Table 5: Scores on GSM8K, MATH, SVAMP, ASDiv. The Accuracy and pass@1 scores of our models are computed with greedy decoding. Models are evaluated in 0-shot on all datasets.

| Model | mode | GSM8K Accuracy(%) | MATH Accuracy(%) | SVAMP Accuracy(%) | ASDiv Accuracy(%) | Avg Accuracy(%) |
|---|---|---|---|---|---|---|
| GPT-4(OpenAI, 2023) | CoT | 92.0 | 42.5 | 93.1 | 91.3 | 79.8 |
| GPT-4(OpenAI, 2023) | Code Interpreter | 94.2 | 51.8 | 94.8 | 92.6 | 83.3 |
| WizardMath-Llama2-7B(Luo et al., 2023a) | CoT | 54.9 | 10.7 | 57.8 | 73.5 | 49.2 |
| MAmmoTH-CodeLlama-7B(Yue et al., 2023) | CoT | 59.4 | 33.4 | 71.4 | 72.3 | 59.1 |
| CodeLlama-7B(Rozière et al., 2023) | Code Interpreter | 34.0 | 16.6 | 59.0 | 61.4 | 42.8 |
| ToRA-CodeLlama-7B(Gou et al., 2023) | Code Interpreter | 72.6 | 44.6 | 70.4 | 78.7 | 66.6 |
| OpenMath-CodeLlama-7B(Toshniwal et al., 2024) | Code Interpreter | 75.9 | 43.6 | 79.6 | 77.7 | 69.2 |
| Qwen-SFT-MoI (ours) | CoT | 74.5 | 31.9 | **84.4** | **88.7** | **69.9** |
| Qwen-SFT-MoI (ours) | Code Interpreter | 67.2 | 31.7 | 82.3 | 84.2 | 66.3 |

Table 6: Evaluation of the MoI training scheme on open-source Llama2 and Llama3 models.

| Model | HumanEval pass@1 | MBPP pass@1 | GSM8K Accuracy(%) | MATH Accuracy(%) | MMLU Accuracy(%) | Avg Accuracy(%) |
|---|---|---|---|---|---|---|
| Llama2-7B-chat(Zhang et al., 2024) | 12.8 | 20.8 | 16.7 | 3.3 | 47.8 | 20.3 |
| Llama3-8B-Instruct(Team et al., 2024) | 62.2 | 70.1 | 79.5 | 30.0 | 67.2 | 61.8 |
| Llama2-7B-MoI (ours) | 61.0 (+48.2) | 52.8 (+32.0) | 74.5 (+57.8) | 30.7 (+27.4) | 52.2(+4.7) | 54.2 (+33.9) |
| Llama3-8B-MoI (ours) | 64.0 (+1.8) | 70.9(+0.8) | 80.2(+0.7) | **35.9**(+5.9) | 67.4(+0.2) | 63.7 (+1.9) |

two-round dialogues. The balance strategy maintains the model's capability in two-round dialogues, though there is a slight performance decline on the HumanEval and MATH datasets. Nonetheless, there is an overall increase in the model's average (Avg) performance metric. The MoI strategy, by integrating chunk-based attention on top of the balance approach, achieves substantial improvements across all tasks.

**Comparison with open-source models.** Table 4 shows that despite the lack of extensive code pre-training, our model, through high-quality SFT and the MoI method, matches the performance of contemporary code models with similar parameters. Comparisons with open-source math reasoning models indicate our model's math skills are on par with those fine-tuned solely on math datasets. Additionally, our model exhibits improved multi-turn dialogue and common sense reasoning capabilities.

**Solving math problems with code interpreter.** We attempted to have the MoI model use a code interpreter for solving mathematical problems. Our evaluation extends beyond MATH and GSM8K datasets to include SVAMP and ASDiv, aiming for a thorough appraisal of mathematical proficiency. Table 5 shows our MoI model, untrained on code interpretation tasks, harnesses integrated Math, Agent, and Coding proficiencies to approach problems. Despite falling short of CoT outcomes, our findings affirm MoI's potential to equip models with a synergistic blend of skills.

Table 7: Comparison of smaller models in the range of 1B to 4B parameters on a variety of benchmarks. MoI tarined models are evaluated in 0-shot.

| Model | HumanEval pass@1 | MBPP pass@1 | GSM8K Accuracy(%) | MATH Accuracy(%) | BBH Accuracy(%) | MMLU Accuracy(%) | Avg Accuracy(%) |
|---|---|---|---|---|---|---|---|
| Tinyllama-1.1B(Zhang et al., 2024) | 6.7 | 19.9 | 2.3 | 0.7 | 28.8 | 24.0 | 13.7 |
| Gemma-2B(Team et al., 2024) | 22.0 | 29.2 | 17.7 | 11.8 | 35.2 | 42.3 | 26.4 |
| StableLM-Zephyr-3B(Bellagente et al., 2024) | 35.4 | 31.9 | 52.5 | 12.5 | 37.7 | 45.9 | 36.0 |
| MiniCPM-2B(Hu et al., 2024) | **50.0** | 47.3 | 53.8 | 10.2 | 36.9 | **53.5** | 42.0 |
| Phi-2(Li et al., 2023b) | 47.6 | **55.0** | 57.2 | 3.5 | **43.4** | 52.7 | 43.2 |
| Qwen1.5-1.8B-chat(Bai et al., 2023) | 20.1 | 18.0 | 38.4 | 10.1 | 24.2 | 46.8 | 26.3 |
| Qwen1.5-4B-chat(Bai et al., 2023) | 25.6 | 29.2 | 57.0 | 10.0 | 32.5 | 56.1 | 31.7 |
| Qwen1.5-1.8B-MoI (ours) | 32.3 (+12.2) | 27.8 (+9.8) | 55.2(+16.8) | 18.3 (+8.2) | 29.8 (+5.6) | 48.1(+1.3) | 34.0 (+7.7) |
| Qwen1.5-4B-MoI (ours) | 40.8(+15.2) | 44.4(+15.2) | **70.0**(+13.0) | **23.8**(+13.8) | 34.7(+2.2) | 58.5(+2.4) | 44.3(+12.6) |

Table 8: Model performance with different attention mask methods. All data underwent balanced sampling and reordering, with the difference being the application of different attention masks. **No attention mask**: The model sees all packed questions and answers during loss computation. **Isolated attention mask**: A strict attention mask and reset position IDs ensure the model sees only individual questions and answers within the packed data. **Chunk-based attention mask**: Following task partitioning and system prompt reordering, internal visibility is allowed among every four instructions (chat, code, math, agent) while each group of four is isolated from the rest using an attention mask.

| Model | HumanEval pass@1 | MBPP pass@1 | GSM8K Accuracy(%) | MATH Accuracy(%) | T-Eval Accuracy(%) | Avg Accuracy(%) |
|---|---|---|---|---|---|---|
| No attention mask | 57.9 | 61.8 | 75.4 | 31.7 | 64.3 | 58.2 |
| Isolated attention mask | 60.3(+2.4) | 54.8(−7.0) | 69.4(−6.0) | 29.5(−2.2) | 69.5(+5.2) | 56.7(−1.5) |
| Chunk-based attention mask | 60.4(+2.5) | 61.0(−0.8) | 74.5(−0.9) | 31.9(+0.2) | 68.2(+3.9) | 59.2(+1.0) |

Table 9: Impact of replacing the weights of different components in language models from a Chat model with those from an SFT model. **Attn** represents the weights of the q, k, v, and o in the attention layer. **Others** represent the weights of the `embedding`, FFN, `Layernorm`, and `lm_head`.

| Qwen Params | | HumanEval pass@1 | | | | GSM8K Accuracy(%) | | | |
|---|---|---|---|---|---|---|---|---|---|
| **Attn** | **Others** | Seq | Packed | Balanced | MoI | Seq | Packed | Balanced | MoI |
| Chat | Chat | 37.2 | 37.2 | 37.2 | 37.2 | 50.3 | 50.3 | 50.3 | 50.3 |
| SFT | SFT | 50.6(+13.4) | 56.1(+18.9) | 55.5(+18.3) | 60.4(+23.2) | 69.2(+18.9) | 71.3(+21.0) | 71.9(+21.6) | 74.5(+24.2) |
| SFT | Chat | 42.7(+5.5) | 42.1(+4.9) | 40.2(+3.0) | 32.9(−4.3) | 62.9(+12.6) | 54.8(+4.5) | 60.2(+9.9) | 47.4(−2.9) |
| Chat | SFT | 48.8(+11.58) | 49.4(+12.19) | 50.6(+13.4) | 44.5(+7.3) | 50.5(+0.2) | 58.6(+8.3) | 44.0(−6.3) | 50.0(−0.3) |

**Effectiveness on small language models and Llama family.** We tested the scalability of the MoI method on smaller models. It can be observed in Table 7 that MoI enhances coding and mathematical capabilities on both 1.8B and 4B version of Qwen1.5. Experimental results on Llama2-7B and Llama3-8B are summarized in Table 6, which demonstrates that MoI could further improves the performance of trained LLMs that were already performing well.

**What is the impact of attention masking in the training of MoI?** We utilized the method outlined in (Dao, 2023) to create a 2D Flash Attention mask, which was then input into the `flash_attn_varlen_func` function to prevent cross-contamination in attention. Table 8 shows that SFT with masking significantly improves reasoning ability on the T-Eval dataset. However, it also leads to a considerable decline in performance on the GSM8K, MATH, and MBPP validation sets. We hypothesize that this issue stems from attention contamination. The final MoI solution, employing a chunk-based attention mask, effectively balances task metrics and mitigates attention contamination.

**Why is MoI effective?** We hypothesize that training with mixed instructions can significantly alter the attention distribution of an SFT model, thereby enhancing its alignment performance. To test this, we conducted a case study by replacing the initial chat model's weights with those from an SFT-trained model, specifically targeting attention and other weights, and observed performance changes on a specific dataset. As shown in Table 9, replacing the chat model's attention weights with MoI's attention weights resulted in a performance decline, a phenomenon not observed in other SFT models. This suggests that the MoI model's attention mechanisms are unable to extract the most relevant knowledge from the original chat MLP. Additionally, the experiments in A.11 also demonstrate the robustness of MoI's attention to irregular system prompts.

## 4 RELATED WORK

### 4.1 LLMS FOR VERSATILE TASKS: MATHEMATICS, CODING, AND TOOL UTILIZATION.

Solving complex mathematical tasks is challenging for open-source LLMs. Research focuses on fine-tuning open-source models (e.g., Touvron et al. (2023a;b); Bai et al. (2023)) with insights from advanced proprietary LLMs (OpenAI, 2023; 2022). WizardMath (Luo et al., 2023a) employs a reinforced evol-instruct method, combining supervised fine-tuning with PPO training to enhance reasoning. Meanwhile, MAmmoTH (Yue et al., 2023) uses Chain of Thought and Program-of-Thought to enable LLMs to work with external tools, like Python interpreters, for mathematical problem-solving.

LLMs have been specifically designed for code generation tasks, such as StarCoder (Li et al., 2023a) and DeepSeek-Coder (Guo et al., 2024). Alternatively, code-generating LLMs can be derived from fine-tuning general-purpose models, exemplified by CodeLlama (Rozière et al., 2023) and WizardCoder(Luo et al., 2023c). Our approach aligns with the latter, fine-tuning general-purpose LLMs to enhance code generation.

LLMs in real-world scenarios must adeptly select and apply tools from numerous APIs. Gorilla (Patil et al., 2023) pairs LLMs with a broad API set, while ToolAlpaca (Tang et al., 2023) documents 3938 tool uses from over 400 real-world APIs in 50 categories. ToolLLM (Qin et al., 2023) further contributes with ToolBench, a rich dataset featuring 16464 real-world APIs across 49 categories from RapidAPI Hub, accommodating diverse single and multi-tool use cases.

### 4.2 MULTI-TASK LEARNING

During multi-task learning, dataset bias (Hadsell et al., 2020) is a phenomenon where two or more tasks pull the model parameters in different directions, thus impacting the multi-task learning performance. GPT-2(Radford et al., 2019) shows language models are unsupervised multitask learners and Flan-T5(Chung et al., 2024) has demonstrated that language models can learn up to 1836 tasks. Liu et al. (2024) proposes concatenating multiple datasets during training and using focal loss to balance data bias among multiple tasks. Dong et al. (2023) proved multi-task learning of LLM lead to conflicts, while sequential training results in catastrophic forgetting. In our work, we propose a method to mitigate dataset bias that focuses on the data rather than the model.

### 4.3 SUPERVISED FINE-TUNING IN THE ALIGNMENT OF LLM

The training process of LLMs is categorized into pretraining and fine-tuning(Touvron et al., 2023b), with Supervised Fine-Tuning (SFT) as key components of the fine-tuning phase. Some researchers, such as (Wei et al., 2021), refer to SFT as instruction-tuning, while Ouyang et al. (2022) defines LLM alignment as 'helpfully and safely following user instructions.' Additionally, studies like (Luo et al., 2023b; Zhang & Wu, 2024; Kotha et al., 2023; Razdaibiedina et al., 2023) have examined catastrophic forgetting during fine-tuning. Furthermore, Wang et al. (2024) explores how incorporating character definitions in system prompts can assist language models in achieving multi-domain alignment while Ge et al. (2024) propose a novel persona-driven data synthesis methodology for SFT. Lu et al. (2024)reveals that the intrinsic capabilities of LLMs confine the knowledge within role-play. In our work, we explored how to efficiently use multiple system prompts to better continue aligning a model that has undergone SFT.

## 5 CONCLUSION

In this work, we highlight the significant role of system prompts in enhancing language model performance during SFT. By refining prompts, we mitigated issues in code generation with the Qwen-7B-chat model, leading to better handling of complex tasks. Our innovative multi-task learning strategy , Mixture of Instructions(MoI), which utilizes curated datasets with tailored prompts, successfully tackled the dataset bias seen in multi-task training. The MoI method further ensured consistent performance across tasks. Our finalized Qwen-SFT-MoI model showcased substantial improvements over the original, underscoring MoI's potential for improving language model adaptability and proficiency.

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

# A APPENDIX

## A.1 DETAILS OF EVALUATION DATASETS

**Mathematic reasoning.** GSM8K (Cobbe et al., 2021) presents diverse grade school math word problems, while the MATH dataset (Hendrycks et al., 2021) focuses on evaluating mathematical reasoning abilities with a collection of problems.In addition, we employed the SVAMP (Patel et al., 2021) and ASDiv (Miao et al., 2021) datasets as supplementary test sets to assess mathematical capabilities.

**Code generation.** HumanEval (Hendrycks et al., 2021) assesses code generation models' problem-solving skills through Python programming challenges, and MBPP (Austin et al., 2021) targets models' execution-based performance on Python problems.

**Tool usage.**T-EVAL (Chen et al., 2023) breaks down the assessment of tool usage into several sub-domains, shedding light on LLMs' comprehensive and specific competencies.

**Common sense and logical reasoning.** MMLU (Hendrycks et al., 2020) covers 57 subjects from STEM to humanities and social sciences, testing models on a wide range of difficulty levels and world knowledge. We utilized the BBH-hard (Suzgun et al., 2022) dataset as an additional test set to evaluate reasoning abilities.

**Multi-turn dialogue.** MT-Bench (Zheng et al., 2024a) features 80 questions across eight domains, including writing, roleplay, reasoning, and more, with each domain offering a two-round query-response evaluation using GPT-4.

## A.2 DETAILS OF EXPERIMENTS

**Qwen-SFT-code.** Trained on MagicCoder110k, CodeFeedback157k, and EvolInstructCode80k.

**Qwen-SFT-math.** Utilized MathInstruct262k and MetaMath395k datasets.

**Qwen-SFT-agent.** Fine-tuned with GPT4Tools75k, AgentInstruct, and ToolbenchG3.

**Qwen-SFT-chat.** Focused on ultrachat200k for dialogue interactions.

**Qwen-SFT-seq.** Combined datasets, following sequence instruction strategy.

**Qwen-SFT-packed.** Merged datasets with packing strategy, learning rate of 4e-5.

**Qwen-SFT-balanced.** Balanced sampling from all datasets, learning rate of 4e-5.

**Qwen-SFT-MoI.** Employed the MoI scheme using "You are a helpful assistant" as the default system prompt, learning rate of 4e-5.

In all experiments, weight decay is set at a factor of 0.1, adam optimizer beta is set to 0.95, Warm-up ratio is set at 0.01, the maximum length of the model is 8192 tokens, gradient checkpointing is enabled.

## A.3 MIXTURE OF INSTRUCTIONS DETAILS

During the training of our language model, we employed the classic ChatML format:

```
<|im_start|>system
You are a helpful assistant.
<|im_end|>
<|im_start|>user
How are you?
<|im_end|>
<|im_start|>assistant
As I am a large language model, I do not have personal emotions.
However, I am functioning properly and ready to assist you with any
information or tasks you need help with. How may I be of service today?
<|im_end|>
```

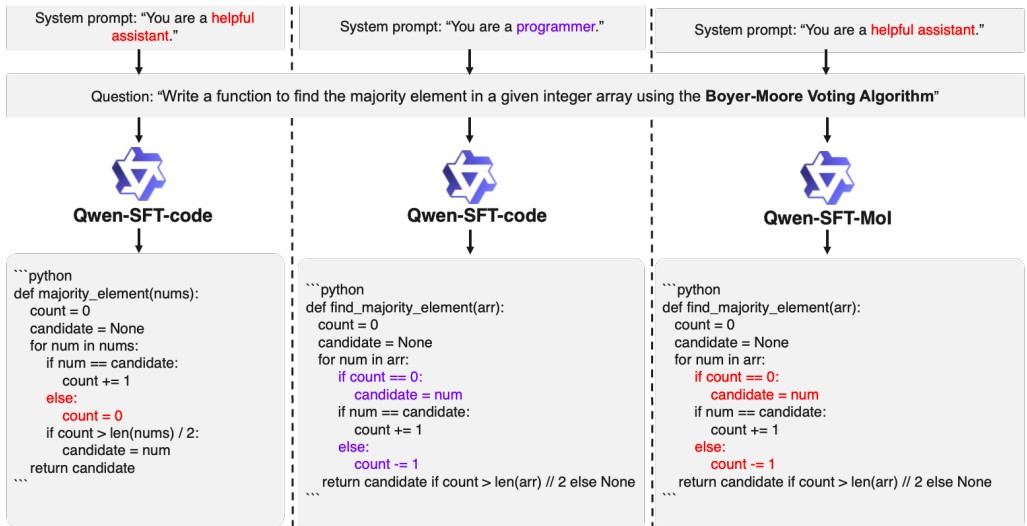

Figure 6: The function of the system prompt and the mapping of knowledge learned from all system prompts onto the default system prompt following the application of the Mixture of Instructions method.

Table 10: Detailed scores of SFT models on T-EVAL. We report zero-shot performance of the models.

| Model | Instruct | Plan | Reason | Retrieve | Understand | Review | overall |
|---|---|---|---|---|---|---|---|
| Qwen-7B-chat | 67.6 | 56.3 | 46.9 | 66.1 | 66.5 | 60.6 | 60.7 |
| Qwen-SFT-code | 0.8 | 0.9 | 35.7 | 29.8 | 52.4 | 14.0 | 22.3 |
| Qwen-SFT-math | 0.3 | 9.2 | 21.5 | 11.2 | 30.3 | 11.7 | 14.0 |
| Qwen-SFT-agent | 44.1 | 22.8 | 37.8 | 58.7 | 63.7 | 55.2 | 47.1 |
| Qwen-SFT-chat | 64.0 | 61.1 | 51.3 | 63.4 | 74.4 | 50.3 | 60.8 |
| Qwen-SFT-seq | **83.7** | **65.8** | **54.4** | **77.1** | **78.4** | **58.3** | **69.6** |
| Qwen-SFT-packed | 83.6 | 62.9 | 51.2 | 72.7 | 75.6 | 56.3 | 67.1 |
| Qwen-SFT-balanced | 82.6 | 64.3 | 51.0 | 73.0 | 75.9 | 54.6 | 66.9 |
| Qwen-SFT-MoI | 78.1 | 65.3 | 53.9 | 76.8 | 77.8 | 57.5 | 68.2 |

In the example above, the portion labeled with 'system' is what we have been discussing as the 'system prompt' in our paper.

Figure 6 illustrates the effects of the Mixture of Instructions (MoI) framework. In the initial phase, models subjected to Supervised Fine-Tuning (SFT) with the default system prompt strive to replicate the canonical implementation of the Boyer-Moore algorithm. However, as the process advances, they divert their approach towards devising an algorithm based on iterative querying. Subsequent to retraining with a modified system prompt, the model acquires the capability to accurately script the Boyer-Moore algorithm, demonstrating a clear understanding of its logic and structure. Utilizing the Mixture of Instructions (MoI) approach for further training enables the seamless transference of knowledge acquired in the *'You are a programmer'* domain to applications within the *'You are a helpful assistant'* scenario, showcasing the versatility and adaptability of the learned skills across different contexts.

### A.4 MIXTURE OF INSTRUCTIONS ATTENTION MAP VISUALIZATION

Figure 2 shows the configuration of the attention map for MoI model, we further visualize the attention maps from each layer within our MoI model during inference in Figure 8, where the attention of each layer is obtained by averaging across all heads' attention in that layers.

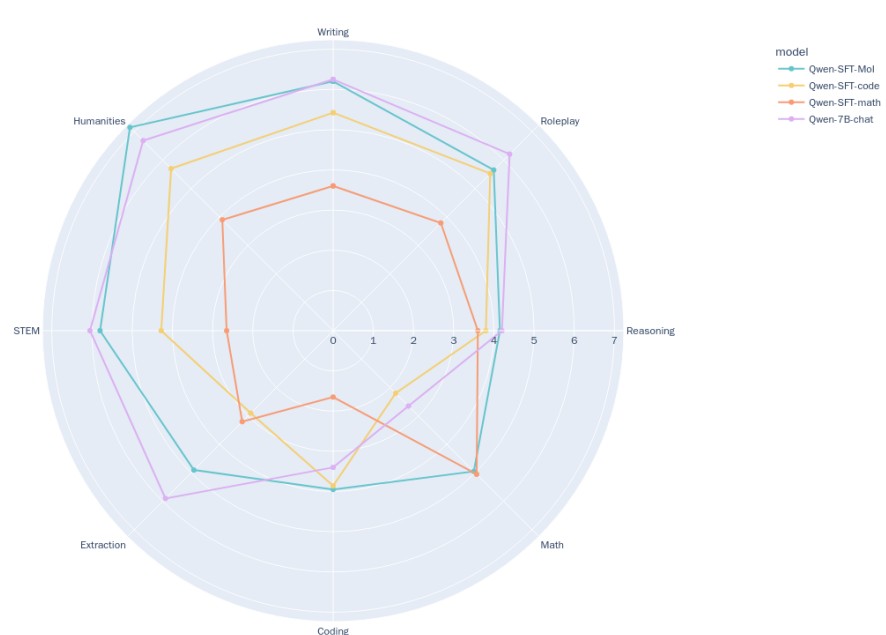

Figure 7: Detailed scoring charts on MT-Bench for Qwen-7B-chat, Qwen-SFT-code, Qwen-SFT-math, and Qwen-SFT-MoI models.

## A.5 MT-BENCH RESULTS

Figure 7 shows that employing the MoI technology in language models concurrently boosts mathematical reasoning and coding skills, paralleling the proficiency achieved by models trained solely on individual tasks. This consistency underscores the robustness of our proposed approach.

## A.6 T-EVAL RESULTS

Table 10 indicates that the agent capabilities within language models exhibit a direct association with their chat functionalities. Subjecting the models to SFT using datasets composed exclusively of code, math, or agent-specific content can significantly impair their agent abilities. The Sequence of Instructions approach outperforms other methodologies in agent-related tasks, which we surmise is due to its compliance with the input format used during the initial SFT, thus maximally preserving the consistency of the language model. Furthermore, our MoI technique secures the highest scores in techniques involving instruction packing, suggesting that our method also aims to retain the original competencies of the language model to the greatest possible extent.

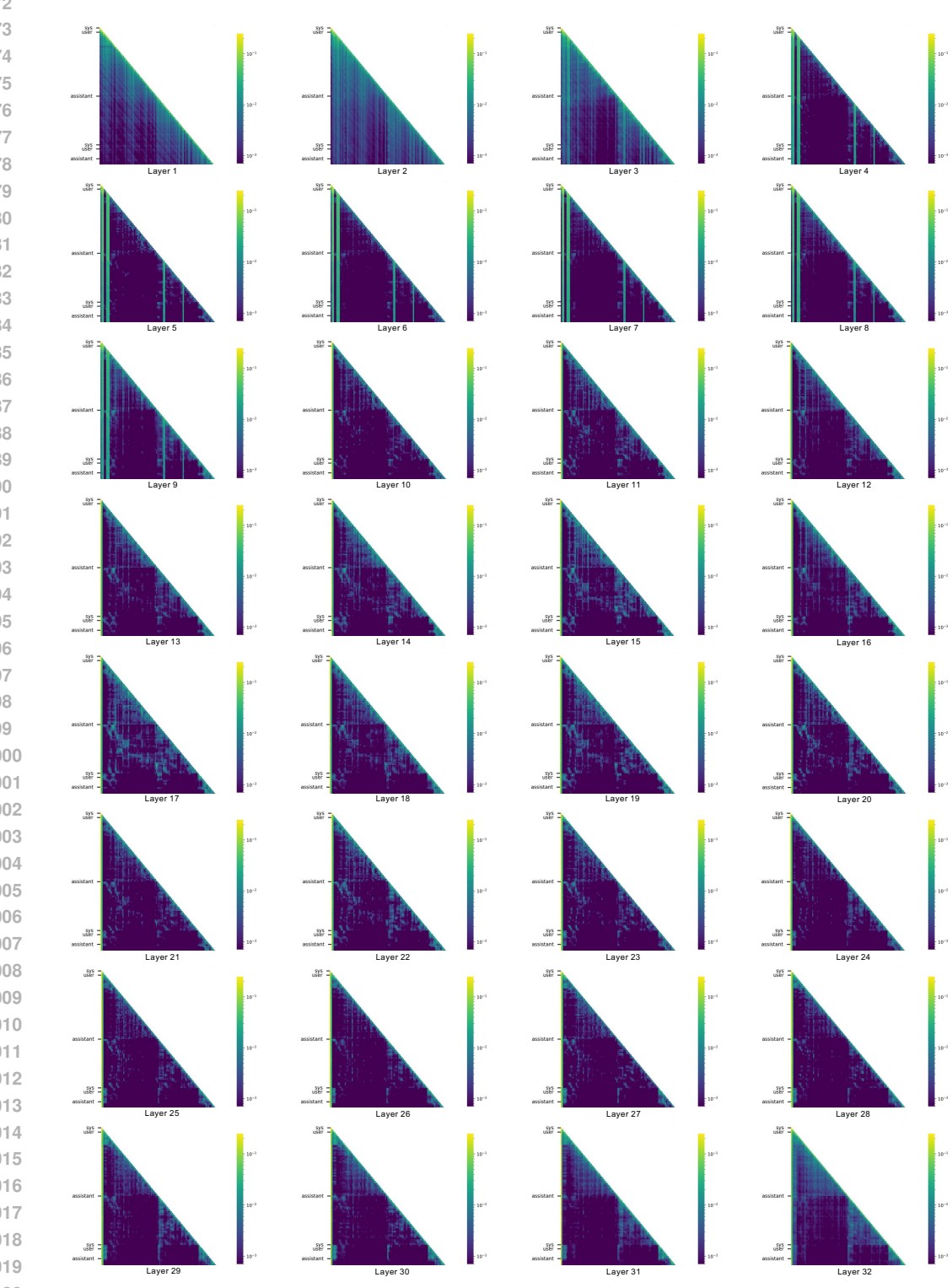

Figure 8: Attention maps from a Qwen-7B-chat model's first, middle, and last layers when processing mixture of instructions. The model attends not only to the current system prompt but also to initial tokens in concatenated instructions. By prefixing mixed instructions with the default system prompt, the model implicitly revisits it during learning of diverse prompts, thereby facilitating cross-domain knowledge transfer to the domain indicated by the default prompt.

## A.7 CASE STUDY

---

**[Turn #1] Benjamin went to a bookstore and purchased a variety of books. He bought 5 copies of a sci-fi novel, each priced at $20, 3 copies of a history book priced at $30 each, and 2 copies of a philosophy book for $45 each. What was the total cost of his purchases?**

Benjamin bought 5 copies of a sci-fi novel, each priced at $20, so the total cost of the sci-fi novels is 5 * $20 = $100. He also bought 3 copies of a history book, each priced at $30, so the total cost of the history books is 3 * $30 = $90. Lastly, he bought 2 copies of a philosophy book, each priced at $45, so the total cost of the philosophy books is 2 * $45 = $90. The total cost of his purchases is $100 + $90 + $90 = $280.
**280**
The answer is 280

**[Turn #2] Suppose Benjamin decides to sell each of these books at a 25% markup from the price he purchased them. What would be his total revenue if he sold all the books he bought?**

Benjamin bought 5 copies of a sci-fi novel, each priced at $20, so the total cost of the sci-fi novels is 5 * $20 = $100. He also bought 3 copies of a history book, each priced at $30, so the total cost of the history books is 3 * $30 = $90. Lastly, he bought 2 copies of a philosophy book, each priced at $45, so the total cost of the philosophy books is 2 * $45 = $90. The total cost of his purchases is $100 + $90 + $90 = $280. If Benjamin decides to sell each of these books at a 25% markup from the price he purchased them, the markup would be 25% of $280, which is 0.25 * $280 = $70. So, the total revenue Benjamin would get if he sold all the books he bought is $280 + $70 = $350.
**350**
The answer is 350

**[Turn #3]After achieving his profit goal by selling books at a markup, Benjamin decides to donate a portion of his total revenue to a local library. If he chooses to donate 15% of his total revenue, how much money will the local library receive?**

Benjamin's total revenue is $350. If he chooses to donate 15% of his total revenue, the amount he will donate is 15/100 * $350 = $52.50. So, the local library will receive $52.50.
**52.5**
The answer is 52.5

**[Turn #4]If Benjamin aims to achieve a profit of $100 over the initial cost of his purchases by selling all the books, what percentage markup should he apply to each book's selling price?**

Benjamin aims to achieve a profit of $100 over the initial cost of his purchases, which is $280. To calculate the markup percentage, we need to find the difference between the selling price and the cost price, and then divide that by the cost price. The selling price is the cost price plus the markup, so the selling price is $280 + $100 = $380. The markup is the difference between the selling price and the cost price, which is $380 - $280 = $100. To find the markup percentage, we divide the markup by the cost price and multiply by 100. The markup percentage is ($100 / $280) * 100 = 35.71%.
**35.71**
The answer is 35.71

---

Table 11: Our MoI model demonstrated performance on the MT-Bench math, and we proceeded to inquire about the third and fourth rounds.

**[Turn #1]Write a simple website in HTML. When a user clicks the button, it shows a random joke from a list of 4 jokes.**

Sure, here is a simple HTML code that does what you asked for:

```
<!DOCTYPE html>
<html>
<head>
   <title>Joke Generator</title>
   <style>
      button {
         margin: 10px;
         padding: 10px;
         font-size: 18px;
      }
   </style>
</head>
<body>
   <h1>Joke Generator>/h1>
   <button onclick="displayJoke()">Generate Joke</button>
   <p id="joke"></p>
   <script >
      const jokes = [
         "Why don't scientists trust atoms? Because they make up everything!",
         "Why did the scarecrow win an award? Because he was outstanding in his field!",
         "Why don't eggs tell jokes? Because they'd crack each other up!",
         "Why don't programmers like nature? Because they prefer to create their own world!"
      ];
      function displayJoke() {
         const randomJoke = Math.floor(Math.random() * jokes.length);
         document.getElementById("joke").innerText = jokes[randomJoke];
      }
   </script >
</body >
</html >
```

This code creates a simple webpage with a button that, when clicked, generates a random joke from the list of jokes and displays it in a paragraph element.

**[Turn #2]How to use CSS to change the color of jokes to red?**

You can use CSS to change the color of the jokes. Here is how you can do it:

```
<!DOCTYPE html>
<html>
<head>
   <title>Random Joke Generator</title>
   <style>
      body {
         font-family: Arial, sans-serif;
      }
      button {
         margin-top: 20px;
      }
      #joke {
         color: read;
      }
   </style>
</head>
<body>
   <h1>Random Joke Generator</h1>
```

```
<button onclick="displayJoke()">Generate Joke</button>
<p id="joke"></p>
<script >
  const jokes = [
    "Why don't scientists trust atoms? Because they make up everything!",
    "Why did the scarecrow win an award? Because he was outstanding in his field!",
    "Why don't eggs tell jokes? Because they'd crack each other up!",
    "Why don't programmers like nature? Because they prefer to create their own world!"
  ];
  function displayJoke() {
    var randomJoke = jokes[Math.floor(Math.random() * jokes.length)];
    document.getElementById("joke").innerHTML = randomJoke;
  }
</script >
</body >
</html >
```

In this code, the CSS rule p { color: red; } is used to change the color of all paragraph elements to red.

**[TURN #3]How to dynamically retrieve a list of jokes from an external file or API**

You can dynamically retrieve a list of jokes from an external file or API using JavaScript's fetch function. Here is an example:

```
<!DOCTYPE html>
<html>
<head>
  <title>Random Joke Generator</title>
  <style>
    body {
      font-family: Arial, sans-serif;
    }
    button {
      margin-top: 20px;
    }
    #joke {
      color: read;
    }
  </style>
</head>
<body>
  <h1>Random Joke Generator</h1>
  <button onclick="displayJoke()">Generate Joke</button>
  <p id="joke"></p>
  <script >
    function displayJoke() {
      fetch('https://your-api-url.com/jokes')
        .then(response =>response.json())
        .then(jokes =>{
          var randomJoke = jokes[Math.floor(Math.random() * jokes.length)];
          document.getElementById("joke").innerHTML = randomJoke;
        })
        .catch(error =>console.error('Error:', error));
    }
  </script >
</body >
</html >
```

Table 12: Our MoI model demonstrated performance on the MT-Bench code, and we proceeded to inquire about the third round.

**Questions:**
You have access to the following API: [
    [
        'AirbnbSearch.get_property_details', {
            'name': 'AirbnbSearch.get_property_details',
            'description': 'This function retrieves the details of a property given its ID.',
            'required_parameters': [
                {'name': 'propertyId','type': 'NUMBER', 'description': 'The ID of the property.'}],
            'optional_parameters': [],
            'return_data': [
                { 'name': 'title', 'description': 'The title of the property.'},
                {'name': 'propertyType', 'description': 'The type of the property.'},
                {'name': 'personCapacity', 'description': 'The capacity of the property.'},
                {'name': 'rating', 'description': 'The rating of the property.'},
                {'name': 'latitude', 'description': 'The latitude of the property.'},
                {'name': 'longitude', 'description': 'The longitude of the property.'}]}
    ],
    [
        'AirbnbSearch.get_property_reviews', {
            'name': 'AirbnbSearch.get_property_reviews',
            'description': 'This function retrieves the reviews of a property given its ID.',
            'required_parameters': [
                {'name': 'propertyId', 'type': 'NUMBER', 'description': 'The ID of the property.'}],
            'optional_parameters': [],
            'return_data': [
                {'name': 'reviews', 'description': 'The reviews of the property, containing comment, rating,
and date. At most 3 reviews are returned.'}]}
    ],
    [
        'AirbnbSearch.check_availability',{
            'name': 'AirbnbSearch.check_availability',
            'description': 'This function checks the availability of a property given its ID.',
            'required_parameters': [
                {'name': 'propertyId', 'type': 'NUMBER', 'description': 'The ID of the property.'}],
            'optional_parameters': [
                {'name': 'date', 'type': 'STRING', 'description': 'The date to check, it must in the format
of YYYY-MM-DD.'}],
            'return_data': [
                {'name': 'availability', 'description': 'The availability of the property for the given date or
the next 10 days.'}]}
    ],
    [
        'AirbnbSearch.search_property_by_place', {
            'name': 'AirbnbSearch.search_property_by_place',
            'description': 'This function takes various parameters to search properties on Airbnb.',
            'required_parameters': [
                {'name': 'place', 'type': 'STRING', 'description': 'The name of the destination.'}],
            'optional_parameters': [],
            'return_data': [
                {'name': 'property', 'description': 'a list of at most 3 properties, containing id, name, and
address.'}]}
    ]
]
Please generate a plan for answer user's questions, which should be a list of actions with the following
format:

```
"""
 [
    {
       // id of the action
       "id": number;
       // the name of the action
       "name": string;
       // input params required by this action
       "args": "str(Record<string, any>)";
    },
    ...
 ]
"""
```

You can imagine args when you plan the action, and these instructions will be executed sequentially.For example, if you want to call 'api1' with 'arg1' and 'arg2', you can write the following plan:

```
"""
 [
    {
       "id": 0,
       "name": "api1",
       "args": "{'arg1': 'value1', 'arg2': 'value2', ...}",
    },
    ...
 ]
"""
```

The args should be a dictionary in string format. PLEASE use '' in args dictionary and use "" in other places, DO NOT print args with value None or null.You should only generate a list in json format. The list should be the full planning list without '...'. DO NOT generate any text to explain the json.
**Here is the instruction:**
I'm helping a client find the first available rental property in Los Angeles. Please find a property in Los Angeles. Then, check the availability of the property for the coming week. Next, get the details of that property. Finally, retrieve the reviews of the property.

**Answer:**
```
 [
    {
       "id": 0,
       "name": "AirbnbSearch.search_property_by_place",
       "args": "{'place': 'Los Angeles'}"
    },
    {
       "id": 1,
       "name": "AirbnbSearch.check_availability",
       "args": "{'propertyId':<property_id>, 'date': '2022-01-01'}"
    },
    {
       "id": 2,
       "name": "AirbnbSearch.get_property_details",
       "args": "{'propertyId': <property_id>}"
    },
    {
       "id": 3,
       "name": "AirbnbSearch.get_property_reviews",
       "args": "{'propertyId': <property_id>}"
    }
 ]
```

Table 13: Our MoI model demonstrated performance on the T-EVAL plan.

## A.8  SOLVE MATH PROBLEM WITH CODE INTERPRETER

In this section, we demonstrate that models trained using the Mixture of Instructions (MoI) approach can perform functions akin to the code interpreter capabilities of GPT-4. This case necessitates a comprehensive evaluation of the model's mathematical reasoning, code writing, and tool utilisation proficiencies.

We feed prompts into the Qwen-SFT-MoI system in the following format, adhering to the ReACT standard during tool use—prompting the model to comply with a Thought, Action, Action Input sequence to generate the tool's output:

```
You are a helpful assistant.
Answer the following questions as best you can. You have access to the
following APIs:

code_interpreter: available for executing Python code. The drive at
'workspace/' can be used to save and persist user files. Internet access
for this session is disabled. Do not make external web requests or API
calls as they will fail.

Parameters:
{
  "type": "object",
  "properties": {
      "code": {"type": "string", "description": "executable code"}
   }
}

Use the following format:
Question: the input question you must answer
Thought: you should always think about what to do
Action: the action to take, should be one of [code_interpreter]
Action Input: the input to the action
Observation: the result of the action
... (this Thought/Action/Action Input/Observation can be
repeated zero or more times)
Thought: I now know the final answer
Final Answer: the final answer to the original input question

Begin!
```

This interaction informs the model to employ a code interpreter tool to address mathematical queries. A model proficient in following instructions should, upon encountering the user's question, compose code that meets the formatting requirements. For this functionality, we have developed an entire pipeline. It filters and extracts the relevant substance from the model's code output, then passes it into a Python code interpreter to obtain execution results, which are subsequently compared with the standard outcomes.

```
Questions:
Henry and 3 of his friends order 7 pizzas for lunch. Each pizza is
cut into 8 slices. If Henry and his friends want to share the pizzas
equally, how many slices can each of them have?

Answer:
Thought:Let's solve this problem using Python code.
Action:code_interpreter
Action Input: {"code":"number_of_pizzas = 7
slices_per_pizza = 8
number_of_friends = 3
number_of_slices_per_person = number_of_pizzas * slices_per_pizza /
```

```
    (number_of_friends + 1)
    number_of_slices_per_person
    "}
    Observation: 14.0
    Thought: I now know the final answer.
    Final Answer: Thus each of them can have \boxed{14} slices.
```

Our model's efficacy is validated against the GSM8K benchmark, where our Qwen-SFT-MoI model achieves an accuracy of 69.14% under a zero-shot setting. Contrastingly, when the model infers separately on the problems, the accuracy reaches 72%. This case illustrates that our model not only possesses commendable mathematical reasoning, code writing, and tool utilisation capacities but can also synergise these abilities to resolve problems encountered in practical processes.

## A.9 THE EFFECTIVENESS OF THE MoI METHOD IN RAG SCENARIOS

Table 14: Detailed scores of SFT models on RGB English Benchmark for Retrieval-Augmented Generation (Chen et al., 2024).

| Model | Noise Ratio 0 | Noise Ratio 0.2 | Noise Ratio 0.4 | Noise Ratio 0.6 | Noise Ratio 0.8 |
|---|---|---|---|---|---|
| ChatGPT(OpenAI, 2022) | 96.33 | 94.67 | 94.00 | 90.00 | 76.00 |
| ChatGLM2-6B(GLM et al., 2024) | 91.33 | 89.67 | 83.00 | 77.33 | 57.33 |
| Vicuna-7B-v1.3(Zheng et al., 2023) | 87.67 | 83.33 | 86.00 | 82.33 | 60.33 |
| Qwen-7B-Chat(Bai et al., 2023) | 94.33 | 91.67 | 91.00 | 87.67 | 73.67 |
| Qwen-SFT-MoI (ours) | 93.33 | 90.00 | 89.67 | 86.33 | 71.67 |

Table 15: Detailed scores of SFT models on L-EVAL(An et al., 2023).

| Model | Coursera | GSM | QuALITY | TOEFL | CodeU | SFiction | Avg |
|---|---|---|---|---|---|---|---|
| ChatGPT(OpenAI, 2022) | 63.51 | 84.00 | 61.38 | 78.43 | 12.22 | 64.84 | 60.73 |
| Llama2-7b-chat(Touvron et al., 2023b) | 29.21 | 19.00 | 37.62 | 51.67 | 1.11 | 60.15 | 33.12 |
| Qwen-7B-Chat(Bai et al., 2023) | 45.64 | 29.0 | 59.40 | 76.20 | 5.55 | 60.93 | 46.12 |
| Qwen-SFT-MoI (ours) | 47.24 | 68.0 | 50.99 | 66.17 | 4.44 | 55.93 | 48.80 |

## A.10 THE EFFECTIVENESS OF THE MoI METHOD ON QWEN 72B WITH LoRA

Table 16: Detailed scores of LoRA(Hu et al., 2021) SFT on Qwen1.5-72B-Chat model. The specific LoRA parameters: r = 32, alpha = 64, learning rate=0.001, . We report zero-shot performance of the models.

| Model | MMLU | GSM8K | MATH | HumanEval | MBPP | Avg |
|---|---|---|---|---|---|---|
| Qwen1.5-72B-Chat(Bai et al., 2023) | 77.5 | 82.7 | 42.5 | 71.3 | 71.9 | 69.18 |
| Qwen1.5-72B-SFT-MoI-LoRA(ours) | 77.9 | 83.0 | 43.1 | 72.8 | 72.2 | 69.80 |

## A.11 CASE STUDY BETWEEN QWEN-7B-CHAT AND QWEN-SFT-MoI

Case 1:

```
System Prompt: You are a helpful assistant. Today is Friday, May 17th.

Question: What day is it today?

Qwen-7B-Chat: Today is Friday, May 17th.

Qwen-SFT-MoI: Today is Friday, May 17th.
```

Case 2:

System Prompt: Today is Friday, May 17th. You are a helpful assistant.

Question: What day is it today?

Qwen-7B-Chat: Today is 17th February 2023.

Qwen-SFT-MoI: Today is Friday, May 17th.

In the two cases above, we set a system prompt for the models. In case 1, we see that both models can respond to questions based on the settings in the system prompt. However, in case 2, we first set the date to Friday, May 17th, and then told the model "You are a helpful assistant." We can observe that the Qwen-7B-Chat model seems to "miss" the time setting, whereas the MoI model can capture this information. This suggests that the MoI model's attention mechanism is more robust to the sequence of input instructions. In contrast, the Qwen-7B-Chat model stubbornly seeks information following the "You are a helpful assistant" keyword. When it cannot find the expected content in this area, the model tends to create an answer rather than capturing information that came before the key setting.

