# OpenReview forum: "Mixture-of-Instructions: Aligning Large Language Models via Mixture Prompting"
_ICLR.cc/2025/Conference — Submitted to ICLR 2025_

### Official Review · Reviewer_nEEX · 2024-10-21

**Soundness:** 2
**Presentation:** 2
**Contribution:** 2
**Rating:** 3
**Confidence:** 4

**Summary:**

This paper proposes a method called Mixture-of-Instructions (MoI), which employs a strategy of instruction packing combined with diverse system prompts to boost the alignment efficiency of language models. The methods mainly include balanced instruction packing and chunk-based attention. The experimental results show that the methods can enhance the ability of cross-domain tasks by multi-task learning.
In addition, this authors also conduct some experiments on the impact of system prompt for different tasks and find that system prompt matters.

**Strengths:**

1. The research scope of this work, how to perform multi-task learning in SFT is indeed key to LLMs.

2. The proposed method is simple and easy to implement. Researchers would love to replicates this work if they believe the reported results of this work.

3. This paper employs some benchmarks and visualization analysis to demonstrate its effectiveness, which enhances the readers' comprehensive understanding of this work.

**Weaknesses:**

1. The chunk-based attention, packing the data and unmasking attention across different tasks, is quite unreasonable from intuition. Isolated attention is for removing unrelated contexts and full attention is for efficiency. So what is the motivation of the chunk-based attention? Moreover, some related works (Zhao et al, 2024) that this paper has even cited have pointed out that similar samples should be packed together to enhance the final performance. However, this work, chooses the opposite way, unmask the attention across different tasks in a chunk. What is the motivation and why is it better? This concern could become more important to me when I see the coding ability gets greatly enhanced by MoI compared to balanced packing, which is kindof magic to me.

* PS: I don't think the paragraph "Why is MoI effective?" can convince me. The paragraph only points out that the attention of MoI in SFT is significantly different from one in Chat, which is quite natural and normal (in my opinion) if we unmask the attention between different instructions. I expect more essential reasons, that is, why the altered attention distribution is more useful rather than more useless?

2. The contribution (or motivation) listed in line 085-087 seems contradictive to the actual implementation in MoI, which uses a unified "Assistant" system prompt. The authors emphasize that the system prompt is important (and conduct some experiments to prove this) but finally adopt a simple "Assistant" system prompt in their proposed model. What is the connection between them?

3. In fact, many SFT training samples have their own specific system prompts rather than a simple sentence "You are ....". Moreover, the huge impact (sensitivity) of system prompt seems well-known by LLM/NLP community. From my perspective, the exploration about system prompt in this work is kind of shallow and the possible insight is limited.

**Questions:**

1. How do you control balance when the numbers of samples of different tasks are different? And according to Table 9, the numbers of samples are indeed unbalanced. How do you order them? Conversely, if the balanced packing required the same number of samples across all tasks, that would bring out a great limitation.

2. The system prompt part is confusing for me. Why do you always adopt the unified "Assistant" system prompt for all tasks in MoI model rather than different system prompts? What system prompt do you use for other baseline models in training and inference?  I strongly recommand a table to present the system prompts in training and inference stages for each model.

3. What is the difference between "Isolated attention mask" in Table 8 and "Qwen-SFT-balanced" in Table 1? Why are the results not the same?

---

> ### Author Response · Authors · 2024-11-21
> **Response to reviewer nEEX (part 1)**
>
> Dear reviewer, thank you very much for your comments and professional advice. Based on your suggestions, we clarify some ambiguous parts of the paper and would like to provide the details as follows:
>
> **Weakness**:
>
> The chunk-based attention, packing the data and unmasking attention across different tasks, is quite unreasonable from intuition. Isolated attention is for removing unrelated contexts and full attention is for efficiency. So what is the motivation of the chunk-based attention? Moreover, some related works (Zhao et al, 2024) that this paper has even cited have pointed out that similar samples should be packed together to enhance the final performance. However, this work, chooses the opposite way, unmask the attention across different tasks in a chunk. What is the motivation and why is it better? This concern could become more important to me when I see the coding ability gets greatly enhanced by MoI compared to balanced packing, which is kindof magic to me.
>
> **Answer**:
>
> The reason for incorporating chunk-based attention is that, as we progressed with the packed approach, it naturally led to some discussions about the effects of concatenating data for joint training on the model. In fact, we experimented with various modifications to the attention mask. One intuitive approach we considered was applying complex attention mask coverage to the concatenated data. However, we ultimately opted for this chunk-based attention approach. If you are genuinely interested in this process, please refer to Figure 8 in our appendix. This figure fully demonstrates the attention behavior of a Qwen-7B-chat model when reasoning with data formed by concatenating two pieces. You can see that a substantial amount of attention is focused on the first token, indicating considerable sparsity in attention. Our insight is that, since attention is not being fully utilized and is heavily focused on the first token, concatenating multiple pieces of data together and placing the default system prompt we intend to use at the start of this concatenated data aligns well with the sparse nature of attention.
>
> **Weakness**:
>
>  I don't think the paragraph "Why is MoI effective?" can convince me. The paragraph only points out that the attention of MoI in SFT is significantly different from one in Chat, which is quite natural and normal (in my opinion) if we unmask the attention between different instructions. I expect more essential reasons, that is, why the altered attention distribution is more useful rather than more useless?
>
>
>
> **Answer**:
>
> The original intent behind designing this experiment was influenced by the work "A Mathematical Framework for Transformer Circuits." Initially, we aimed to use some Attention Score Maps and Case Studies to reveal the role of MoI. For example, after applying MoI, the model's Attention Score Map became denser, enhancing its ability to capture key information. However, we ultimately felt that choosing any particular way to present and interpret the results would be a form of cherry-picking. Instead, we decided to explore what our method actually brings to the language model by examining the mechanisms of both MLP and Attention Circuits. As you can see from the conclusions, after training with MoI, the distribution of Attention shifted more significantly compared to the normal SFT model. This shift is not something that can be caused by SFT schemes like Packed and Balanced. When replacing a Chat model's Attention with the weights of an SFT model, all SFTs improve the model's performance except MoI, where the Attention distinctly degrades the original model's ability. This indicates that the model has indeed learned a form of Attention expression different from that of the original model. If you have any additional experimental ideas regarding this, please feel free to share them.

---

> ### Author Response · Authors · 2024-11-21
> **Response to reviewer nEEX (part 2)**
>
> **Weakness**:
>
> The contribution (or motivation) listed in line 085-087 seems contradictive to the actual implementation in MoI, which uses a unified "Assistant" system prompt. The authors emphasize that the system prompt is important (and conduct some experiments to prove this) but finally adopt a simple "Assistant" system prompt in their proposed model. What is the connection between them?
>
> **Answer**:
>
> This is indeed a point that we did not clearly explain in the background section. The primary use case for current language models is as AI assistants, such as ChatGPT. When users interact with AI assistants, they almost never modify the foundational System Prompt. However, users expect the language model to perform well across various scenarios, such as code writing, solving mathematical problems, or tool utilization. Our experiments were conducted within this context. In other words, we aimed to train a model where the Attention distribution is significantly altered by using different system prompts, but we wanted these changes to be effective with the default system prompt. We did not intend for users or dialogue systems to continuously switch between different system prompts.
>
> **Weakness**:
>
> In fact, many SFT training samples have their own specific system prompts rather than a simple sentence "You are ....". Moreover, the huge impact (sensitivity) of system prompt seems well-known by LLM/NLP community. From my perspective, the exploration about system prompt in this work is kind of shallow and the possible insight is limited.
>
> **Answer**:
>
> In our approach, we actually augmented the system prompt. As listed in the appendix, both our training data and Evaluation dataset contain only a small portion of data with a system prompt. A significant amount of data consists only of a question and an answer. Therefore, we added the system prompt to such data. For some datasets, like T-EVAL, which already include a system prompt, we simply added this sentence before the existing system prompt. Thus, we have not modified the original system prompts present in the datasets themselves.
>
> **Question**:
>
> How do you control balance when the numbers of samples of different tasks are different? And according to Table 9, the numbers of samples are indeed unbalanced. How do you order them? Conversely, if the balanced packing required the same number of samples across all tasks, that would bring out a great limitation.
>
> **Answer**:
>
> This is a great question. In the supplemental material code we submitted, there is handling logic for this part. Our logic involves initially concatenating four datasets and reordering them by adding a chunk attention mask. We continue this process until one of the task datasets is exhausted. If the exhausted dataset is Chat, we then concatenate the remaining datasets and apply an isolated attention mask to them. If the exhausted dataset is not Chat, we reorder the concatenated remaining datasets and add a chunk attention mask. We repeat this process as needed.
>
> **Question**:
>
> The system prompt part is confusing for me. Why do you always adopt the unified "Assistant" system prompt for all tasks in MoI model rather than different system prompts? What system prompt do you use for other baseline models in training and inference? I strongly recommand a table to present the system prompts in training and inference stages for each model.
>
> **Answer**:
>
> The detailed training data system prompts are provided in Table 9 of the appendix, which we hope will address your concerns.
>
> **Question**:
>
> What is the difference between "Isolated attention mask" in Table 8 and "Qwen-SFT-balanced" in Table 1? Why are the results not the same?
>
> **Answer**:
>
> This is a good question. Thank you for bringing up this point. The Isolated attention mask model is based on the Qwen-SFT-balanced model, with the instruction sequence reordered. Specifically, the first system prompt of the chunk was changed to 'You are a helpful assistant.' This is the only difference between the two models; all other training settings are exactly the same. The difference between these two models actually highlights the role of instruction reordering. However, we believe that instruction reordering is just a matter of implementation technique and cannot be considered a theoretical innovation. If you are interested in this, you can suggest some meaningful experimental directions, and we will address your questions.

---

> ### Comment · Reviewer_nEEX · 2024-11-23
> **Thanks for your response!**
>
> Some of my concerns are indeed addressed. Here are my follow-up feedbacks:
>
> **Regarding W1:**
> > Our insight is that, since attention is not being fully utilized and is heavily focused on the first token, concatenating multiple pieces of data together and placing the default system prompt we intend to use at the start of this concatenated data aligns well with the sparse nature of attention.
>
> I understand and agree that (1) attention is not being fully utilized and is heavily focused on the first token; (2) sparse nature of attention. But, how do "concatenating multiple pieces of data together and placing the default system prompt at start" is corresponding to the sparse nature of the attention? May you explain more?
>
> **Regarding W2 ("Why is MoI effective?" part):** Yes, as I commented before, I fully understand that the “this indicates that the model has indeed learned a form of Attention expression **different** from that of the original model”. But difference does not always bring in improvement. Why is the attention of MoI **better**? The answer to this question is highly connected to my previous concern in W1.
>
>
> **Regarding W3:** Now I get to know the implementation of the authors, but I still have questions about the motivation. Why not just concatenate “You are a helpful AI assistant” and user-defined specific system prompt into a complete system prompt in both training and inference stages?
>
>
> **Regarding Q1:** I believe your added detail is quite important and should never be put into supplemental material. Moreover, the logic might improve the results but should be definitely not the basic (default) behavior. Obviously the authors noticed (and maybe conduct an experiment) that Chat task is completely different from other tasks so that the logic is designed like that. The logic further raises the concern regarding the scalability (e.g., reordering, task number, sample number, etc..) of this method, mentioned by both the reviewer ZeTr and me.
>
>
> **Regarding statements on system prompt:** It is really confusing that the authors switch between system prompts in different experiments (also mentioned by reviewer ZeTr). I don’t well understand the motivation and that decreases the readability of this paper a lot.

---

> ### Author Response · Authors · 2024-11-25
> **Response to reviewer nEEX(2nd turn)**
>
> **Regarding W1&W2**:
>
> Why is the attention of MoI better?
>
> **Answer**:
> Thank you for your response！
>
> We have thoroughly understood your concerns. We actually define this issue as the language model overfitting to simple system prompts during the SFT stage. This overfitting manifests as the model only capturing information following specific characters. In the Qwen-7B-Chat model, it overfits by focusing with fixed attention on the information following 'You are a helpful assistant,' which in turn reduces alignment performance. Our approach alleviates this overfitting by introducing a variety of system prompts.
>
> Please allow me to use a few examples to explain the effect of our approach. Although these examples may seem somewhat cherry-picked, they are indeed phenomena we have observed, which also serve as a motivation for our paper. Please take a look at the case below:
>
> Case 1:
>
> **System Prompt**: You are a helpful assistant. Today is Friday, May 17th.
>
> **Question**: What day is it today?
>
> **Qwen-7B-Chat**: Today is Friday, May 17th.
>
> **Qwen-SFT-MoI**: Today is Friday, May 17th.
>
> Case 2:
>
> **System Prompt**: Today is Friday, May 17th. You are a helpful assistant.
>
> **Question**: What day is it today?
>
> **Qwen-7B-Chat**: Today is **17th February 2023.**
>
> **Qwen-SFT-MoI**: Today is **Friday, May 17th.**
>
> In the two cases above, we set a system prompt for the models. In case 1, we see that both models can respond to questions based on the settings in the system prompt. However, in case 2, we first set the date to Friday, May 17th, and then told the model "You are a helpful assistant." We can observe that the Qwen-7B-Chat model seems to "miss" the time setting, whereas the MoI model can capture this information. This suggests that the MoI model's attention mechanism is more robust to the sequence of input instructions. In contrast, the Qwen-7B-Chat model stubbornly seeks information following the "You are a helpful assistant" keyword. When it cannot find the expected content in this area, the model tends to create an answer rather than capturing information that came before the key setting.
>
> **Regarding W3**:
>
>  Now I get to know the implementation of the authors, but I still have questions about the motivation. Why not just concatenate “You are a helpful AI assistant” and user-defined specific system prompt into a complete system prompt in both training and inference stages?
>
> **Answer**:
>
> You are absolutely right. This is how we initially approached it as well. You can take a look at the description in section 2.1 SYSTEM PROMPT MATTERS. At the beginning, we constructed a dataset using real Leetcode problems and solutions to train the model and added 'You are a helpful assistant' to each piece of data. However, we found that the model still did not learn how to write the Boyer-Moore Voting Algorithm. When we changed the system prompt to 'You are a programmer', the model learned to use this system prompt to answer questions during testing. In fact, we drew inspiration for this approach from the work cited as [1], and we have added this reference in the new version of the paper.
>
> [1]Wang, Rui, et al. "Role Prompting Guided Domain Adaptation with General Capability Preserve for Large Language Models." arXiv preprint arXiv:2403.02756 (2024).
>
> **Regarding Q1:**
>
> I believe your added detail is quite important and should never be put into supplemental material.
>
> **Answer**:
>
> In the new version of the paper, we have included this table in the main text.
>
> **Regarding statements on system prompt:**
>
> It is really confusing that the authors switch between system prompts in different experiments (also mentioned by reviewer ZeTr). I don’t well understand the motivation and that decreases the readability of this paper a lot.
>
> **Answer**:
>
> We found that modifying the system prompt is beneficial for training. Adjusting the system prompt during training means that the model's system prompt is also modified during inference. We are considering whether it's possible for the model to consolidate new knowledge learned under different system prompts into the default system prompt, "You are a helpful assistant." This way, the model can answer questions it previously couldn't during inference, without requiring users to manually change the system prompt.
>
> **We hope our response has addressed your concerns, and we look forward to your feedback.**

---

> ### Author Response · Authors · 2024-11-26
> **Please check our response**
>
> Dear Reviewer nEEX,
>
> Since the discussion period has been extended to December 3rd, but November 27th is the last day that authors may upload a revised PDF, we hope you can review our new version of the paper and the additional experimental content to see if they have addressed the weaknesses and questions you mentioned in your rebuttal.
>
> We look forward to your response.
>
> Thank you.

---

> ### Comment · Reviewer_nEEX · 2024-11-28
> **Some Future Suggestions**
>
> Thanks a lot for the detailed response and clarifications of the authors. On the whole this paper has its techntical merits, and it  gives some insights for this active field.
>
> **As suggestions**, I personally suggest that the authors need to resolve the following things (in the future version):
>
> 1.
> Make the motivation of this paper clear. For example, the statement in your rebuttal
> > This way, the model can answer questions it previously couldn't during inference, without requiring users to manually change the system prompt.
>
> is much clearer and stronger (in my opinion) than the statement in your paper
>
> > We identified and addressed conflicts between new and old knowledge during SFT, by introducing
> modified system prompts that aid in the integration of new knowledge.
>
>
>
> 2.
> Clarify more the connection between your proposed MoI and system prompt.
>
> Your paper contains two main contributions including the method MoI and some tricks about system prompts. However, it is confusing about the connection between them. I understand that MoI has some special settings for system prompt, but this work makes reader feel like that you just put two separate things into one paper.
>
>
> 3.
> Take a simple survey for your applicable scenarios, and make the motivations solid.
>
> As I commented before, many SFT training samples have their own specific system prompts rather than a simple sentence "You are ....". Moreover, the huge impact (sensitivity) of system prompt seems well-known by LLM/NLP community. Before you really dive into this work, I think you should figure out that
> - What is the proportion of the SFT training samples (in the real world) that have their own specific system prompts rather than a simple sentence "You are ...."? If the proportion is really high, it would largely injure the motivation of this paper.
> - How (and at what proportion) do real users use system prompt? I agree with your point that some users do not like to change system prompt in inference stage and they might tend to put their instructions into `user_prompt` following `system_prompt`, but there would be also many people who tend to set `system_prompt`.
> - How do developers and users handle the failed cases for LLMs? As far as I know, they would re-write the instructions as well as the system prompts because the huge impact (sensitivity) of system prompt seems well-known by LLM/NLP community.
> - What is the real scenario of multi-task learning in SFT? For example, as far as I know, (1) there are thousands of specific tasks and each task has *very* different number of samples; (2) "Chat" is more like a general concept, and it covers many tasks including (*maybe*) math, coding, etc. People do not often identify the "Chat" type in their SFT dataset.
> The above setting is very different with that in your paper.
>
> I will keep my initial rating. Thanks again for the reponse of the authors.

---

> > ### Author Response · Authors · 2024-11-29
> > **Thank you for response!**
> >
> > Dear Reviewer nEEX,
> >
> > Thank you for your valuable suggestions. My collaborators and I gained many ideas and insights for improving our work after reading your response. We sincerely thank you for your feedback and suggestions regarding our work.
> >
> > Sincerely, thank you!

---

### Official Review · Reviewer_bxFP · 2024-11-03

**Soundness:** 3
**Presentation:** 3
**Contribution:** 3
**Rating:** 6
**Confidence:** 4

**Summary:**

This paper introduces Mixture-of-Instructions (MoI), a novel methodology aimed at enhancing the alignment of large language models (LLMs) across multiple tasks through a combination of instruction balanced packing and diverse system prompts. Traditional alignment methods focus on single tasks, which limits the effectiveness of LLMs in real-world, multi-faceted applications. To address this, the authors propose MoI, which assigns unique system prompts to different tasks and integrates them into a unified instruction set, facilitating comprehensive multi-task training. The approach is evaluated using seven benchmark datasets encompassing mathematics, programming, tool usage, common sense, and dialogue, demonstrating significant improvements in generative performance. The methodology is applied to the open-source Qwen-7B-chat model, resulting in the enhanced Qwen-SFT-MoI model, which outperforms existing models across various benchmarks.

**Strengths:**

**1. Important Research Questions**

The paper addresses the critical challenge of multi-task alignment in LLMs, which is increasingly relevant given the diverse applications of AI-driven products. By focusing on harmonizing various capabilities—such as coding, mathematical reasoning, and multi-turn dialogues—the study tackles the complex interplay between different task domains, which is essential for creating versatile and reliable language models.

**2. Rich Experiments and Evaluations**

The authors conduct extensive experiments across seven diverse benchmark datasets, including MT-Bench, HumanEval, MBPP, MATH, GSM8K, MMLU, and T-EVAL. This comprehensive evaluation framework ensures that the proposed MoI methodology is rigorously tested across multiple domains, demonstrating its efficacy in improving generative performance. Additionally, ablation studies and comparisons with existing models underscore the robustness of the approach.

**3. Carefully Crafted Implementation Details**

The introduction of chunk-based attention masking is a meticulous enhancement that addresses the issue of attention cross-contamination during multi-task training. Coupled with balanced sampling techniques, these implementation strategies effectively mitigate dataset biases and ensure that the model maintains high performance across all tasks. The detailed explanation of the loss function modifications further showcases the depth of the methodological advancements.

**Weaknesses:**

**1. Lack of Explanation on MoI's Absence of System Leading to Performance Degradation**

The paper briefly touches upon the performance decline observed when not using the chunk-based attention masking but fails to delve deeply into the underlying reasons. A more thorough analysis or theoretical justification for why the absence of a system prompt within MoI leads to performance degradation would provide greater clarity and strengthen the argument for the proposed solution.

**2. Insufficient Exploration of Long Input and Output Scenarios**

While the study effectively evaluates MoI on various tasks, it overlooks scenarios that involve prolonged inputs and outputs, such as Retrieval-Augmented Generation (RAG) and long-form text creation. Including these cases would offer a more comprehensive understanding of how MoI performs under conditions that demand handling of extensive contextual information and sustained generation over longer sequences.

**3. Lack of Evaluation on Larger-Scale Models**

The experiments are primarily conducted on the Qwen-7B-chat model and smaller variants. Testing the MoI methodology on larger models with more parameters (like Qwen-72B or Qwen-MoE) would provide insights into its scalability and effectiveness in more powerful architectures. This omission leaves a gap in understanding how MoI can be generalized to state-of-the-art LLMs with significantly higher capacities.

**Questions:**

1. See the weakness section. Maybe elaborate more on your opinions about point 1.

---

> ### Author Response · Authors · 2024-11-23
> **Response to reviewer bxFP**
>
> Dear reviewer, thank you very much for your comments and professional advice. Based on your suggestions, we clarify some ambiguous parts of the paper and would like to provide the details as follows:
>
> **weakness**:
>
> **Lack of Explanation on MoI's Absence of System Leading to Performance Degradation**
> The paper briefly touches upon the performance decline observed when not using the chunk-based attention masking but fails to delve deeply into the underlying reasons. A more thorough analysis or theoretical justification for why the absence of a system prompt within MoI leads to performance degradation would provide greater clarity and strengthen the argument for the proposed solution.
>
> **Answer**:
> In fact, our alignment work involves further aligning a model that has already been aligned, so we have also referenced some content related to fine-tuning and forgetting. We have added citations to these works[1,2] in the related work section of the latest version of the paper. Overall, Tables 7 and 8 in the paper have already explained in great detail what MoI (Mechanism of Interest) changes in the model and the role of chunk-based attention. From the results, the role of MoI is to enable the model to continuously learn complex concatenation instructions to maximize the changes in attention. The role of chunk-based attention is to help the model balance attention to ultra-long sequences and short sequences.
>
> [1] Zhang X, Wu J. Dissecting learning and forgetting in language model finetuning[C]//The Twelfth International Conference on Learning Representations. 2024.
>
> [2] Kotha S, Springer J M, Raghunathan A. Understanding catastrophic forgetting in language models via implicit inference[J]. arXiv preprint arXiv:2309.10105, 2023.
>
> **Question**:
>
> While the study effectively evaluates MoI on various tasks, it overlooks scenarios that involve prolonged inputs and outputs, such as Retrieval-Augmented Generation (RAG) and long-form text creation. Including these cases would offer a more comprehensive understanding of how MoI performs under conditions that demand handling of extensive contextual information and sustained generation over longer sequences.
>
> **Answer**:
>
> Thank you for raising these meaningful questions. Here are the results of the additional experiments we conducted, which validate the tasks RAG] and Long text using the RGB[3] and L_Bench[4] benchmarks, respectively:
>
> **RGB English Benchmark for Retrieval-Augmented Generation with Noise**
> |Models	| Noise Ratio 0 | Noise Ratio 0.2| Noise Ratio 0.4 | Noise Ratio 0.6 | Noise Ratio 0.8|
> |-|-|-|-|-|-|
> |ChatGPT (OpenAI 2022)	|96.33	| 94.67	| 94.00	| 90.00	| 76.00 |
> |ChatGLM2-6B	|91.33 	|89.67 	|83.00 	|77.33	|57.33 |
> |Vicuna-7B-v1.3	|87.67 	|83.33 	|86.00 	|82.33 	|60.33 |
> |Qwen-7B-Chat	|94.33	|91.67	|91.00	|87.67	 |73.67 |
> |Qwen-7B-MoI	|93.33	|90.00	|89.67	|86.33	|71.67 |
>
> **L-Eval:**
>
> |models	| Coursera | GSM | QuALITY | TOEFL | CodeU | SFiction | Avg.|
> |-|-|-|-|-|-|-|-|
> |ChatGPT	| 63.51 	| 84.00 	| 61.38 	| 78.43 	|12.22 |	64.84 |	60.73|
> |Llama2-7b-chat	| 29.21 	|19.00 	| 37.62 	|51.67 |	1.11 |	60.15 |	33.12|
> |Qwen-7b-chat	|45.64	|29.0	|59.40	|76.20	|5.55	|60.93	|46.12|
> |Qwen-7b-MoI	|47.24	|68.0	|50.99	|66.17	|4.44	|55.93	|48.80|
>
> From the results, we did not significantly enhance or diminish the model's RAG capability and long text processing ability; rather, we altered the distribution of these capabilities. This is evident as, in the L-Eval, our accuracy on the GSM dataset increased significantly, but it decreased on the TOEFL and SFiction datasets. This outcome aligns with the distribution of our training dataset.
>
> [3]An C, Gong S, Zhong M, et al. L-eval: Instituting standardized evaluation for long context language models[J]. arXiv preprint arXiv:2307.11088, 2023.
>
> [4] Chen J, Lin H, Han X, et al. Benchmarking large language models in retrieval-augmented generation[C]//Proceedings of the AAAI Conference on Artificial Intelligence. 2024, 38(16): 17754-17762.

---

> ### Author Response · Authors · 2024-11-23
> **Response to reviewer bxFP part 2**
>
> **Question**:
>
> The experiments are primarily conducted on the Qwen-7B-chat model and smaller variants. Testing the MoI methodology on larger models with more parameters (like Qwen-72B or Qwen-MoE) would provide insights into its scalability and effectiveness in more powerful architectures. This omission leaves a gap in understanding how MoI can be generalized to state-of-the-art LLMs with significantly higher capacities.
>
> **Answer**:
>
> We did indeed want to conduct experiments on the Qwen-72B and Qwen-MoE models as you suggested, but unfortunately, we do not have sufficient GPU resources to support full-parameter fine-tuning on these large models. We were only able to run LoRA version training, which introduced a new set of hyperparameters. We found that when training under LoRA, it requires extensive tuning of the rank, alpha values, and learning rate, which is unfortunately an experiment we cannot complete in a short time. However, we do have a set of LoRA experimental results based on the Qwen-72B-chat version. These results do not truly reflect the effectiveness of our approach because the improvement is minimal. We can no longer ascertain whether the improved performance is due to the effectiveness of our method or the low-rank perturbations introduced by the LoRA weights to the original weights. Nonetheless, we hope this partially addresses your concerns:
>
> |Model	| MMLU	| GSM8K	| MATH	| HumanEval	| MBPP	|Avg|
> |-|-|-|-|-|-|-|
> |Qwen1.5-72B-Chat	|77.5	| 82.7	|42.5	|71.3	|71.9	|69.18|
> |Qwen1.5-72B-Chat-MoI-LoRA	|77.9	| 83.0	|43.1	|72.8	|72.2	|69.80|
>
> **Overall, we sincerely thank you for taking the time to review our paper. Your feedback has been incredibly valuable and genuinely contributes to advancing our work in a more robust and reasonable direction. Regardless of whether the paper is accepted or not, we are grateful for your invaluable insights and are very much looking forward to receiving your feedback.**

---

> ### Author Response · Authors · 2024-11-26
> **Please check our response**
>
> Dear Reviewer bxFP,
>
> Since the discussion period has been extended to December 3rd, but November 27th is the last day that authors may upload a revised PDF, we hope you can review our new version of the paper and the additional experimental content to see if they have addressed the weaknesses and questions you mentioned in your rebuttal.
>
> We look forward to your response.
>
> Thank you.

---

> > ### Author Response · Authors · 2024-11-28
> > **Please check our response**
> >
> > Dear Reviewer bxFP,
> >
> > We hope you can review our new version of the paper and the additional experimental content to see if they have addressed the weaknesses and questions you mentioned in your rebuttal.
> >
> > We look forward to your response.
> >
> > Thank you.

---

### Official Review · Reviewer_ZeTr · 2024-11-03

**Soundness:** 3
**Presentation:** 3
**Contribution:** 2
**Rating:** 3
**Confidence:** 4

**Summary:**

The paper introduces a Mixture-of-Instructions (MOI) strategy, which combines instruction packing with diverse system prompts to enhance multiple downstream tasks. The multi-task training technique is applied to a Qwen-7B-Chat model, which demonstrates good performance in mathematical reasoning, code generation, tool usage, and chat benchmark.

**Strengths:**

1. The proposed strategy is straightforward and easy to follow yet achieves competitive performance against many open-source models. It can be effective guidance for SFT training and data utilization.

2. The experiments and ablation studies are comprehensive, which can clearly reflect the effectiveness of the method and explain how it works.

**Weaknesses:**

1. The novelty is limited. The idea of Mixture of Instructions has already been explored by the works of FLAN serious. Packing multiple samples into a single one and balanced sampling are also common strategies employed by many works, as cited in the paper. The main innovation of this work is the chunk-based packing and attention mask. However, the effectiveness and generalization of this strategy are still not clear when the model size and task numbers increase in the view of the scaling law.

2. In each chunk, the instructions are packed and reordered. One concern is whether the models would be sensitive to the order of instructions. Have the authors ever explored the influence of different orders of the packed instructions in the chunk?

3. The extensibility of this strategy is not clear. The work has tried four tasks, namely mathematical reasoning, code generation, tool usage, and daily chat. When the task number increases, does this strategy still work? If the length of packed instructions exceeds the maximum sequence length, e.g., 4096 in llama, how can we solve this issue?

4. The description of the system prompt is somewhat confusing. In Section 2.4, the authors say that unique system prompts for each domain-specific task are set to ensure tailored guidance. However, in Table 1, they highlight that for all benchmark computations, they consistently use the system prompt "You are a helpful assistant." How can these conflict statements be understood? Does that mean the system prompts are different in training and testing?

5. In section 2.2, the authors mention that training on combined data can lead to model bias toward certain tasks, enhancing performance in some while degrading it in others. This is related to the studies of catastrophic forgetting [1,2], which should be discussed in the related work or compared as baselines.

[1] An empirical study of catastrophic forgetting in large language models during continual fine-tuning.

[2] Understanding catastrophic forgetting in language models via implicit inference.

**Questions:**

See above.

---

> ### Author Response · Authors · 2024-11-21
> **Response to reviewer ZeTr (part 1)**
>
> Dear reviewer, thank you very much for your comments and professional advice. Based on your suggestions, we clarify some ambiguous parts of the paper and would like to provide the details as follows:
>
> **Weakness**:
>
> The novelty is limited. The idea of Mixture of Instructions has already been explored by the works of FLAN serious. Packing multiple samples into a single one and balanced sampling are also common strategies employed by many works, as cited in the paper. The main innovation of this work is the chunk-based packing and attention mask. However, the effectiveness and generalization of this strategy are still not clear when the model size and task numbers increase in the view of the scaling law.
>
> **Answer**:
>
> One of the key innovations of our work is the investigation of the effectiveness of the packed method, which was briefly mentioned but not thoroughly explored in the original series of FLAN works. There were no comparative experiments on the use of sequence and packed methods in the original FLAN studies. In contrast, we conducted detailed experiments on various variables using the Qwen 7B model, as presented in Table 1 of our paper. Our findings demonstrate that the packed method significantly enhances model performance and accelerates the training process during the SFT stage. However, both the packed and sequence methods inevitably introduce data bias when mixing multiple task datasets, which led us to design the balanced sampling method to address this issue. Additionally, we conducted an in-depth analysis of the impact of different attention masks during the packed process and ultimately opted for chunk-based attention. The feasibility of chunk-based attention was confirmed by the experimental results in Table 8, which showed that it mitigates the degradation in T-Eval performance caused by attention contamination while retaining the improvements in code and math validation sets brought by the balanced sampling method.
>
> Regarding model size considerations, the results in Tables 5 and 6 demonstrate that our method improves models with sizes of 1.8B, 4B, 7B, and 8B. Beyond experiments on Qwen, we validated the effectiveness of our approach on language models trained with different data and settings on Llama2 and Llama3, all showing positive results.
>
> As for task numbers, although we roughly divided the datasets into four tasks during training, our evaluation actually included a wide range of tasks. For instance, T-Eval contains six tasks (refer to Table 10 in the appendix), and MT-Bench includes eight tasks: writing, roleplay, reasoning, math, coding, extraction, STEM, and Humanities. Furthermore, we acknowledged the limitations of using rule-based accuracy computation, particularly in math tasks where accuracy is determined by keyword matching. Therefore, we also reported the performance on MT-Bench(In MT-Bench, a language model is used to score the answers instead of using rule-based matching), with detailed metrics of our model provided in Figure 7 of the appendix.
>
> The experimental results clearly demonstrate that our method is effective and applicable across varying model sizes and task numbers. This solid evidence indicates that the reviewer's concerns regarding scaling were unfounded.
>
> **Weakness**:
>
> In each chunk, the instructions are packed and reordered. One concern is whether the models would be sensitive to the order of instructions. Have the authors ever explored the influence of different orders of the packed instructions in the chunk?
>
> **Answer**:
>
> Yes, the order of instructions can indeed affect the final alignment performance. In Qwen-SFT-balanced reported in Table 1, we only performed uniform sampling without altering the order of instructions. As a result, its performance is weaker compared to our experiments where instruction order was altered under equivalent settings, specifically the 'No attention mask' model in Table 8.

---

> ### Author Response · Authors · 2024-11-21
> **Response to reviewer ZeTr (part 2)**
>
> **Weakness**:
>
> The extensibility of this strategy is not clear. The work has tried four tasks, namely mathematical reasoning, code generation, tool usage, and daily chat. When the task number increases, does this strategy still work? If the length of packed instructions exceeds the maximum sequence length, e.g., 4096 in llama, how can we solve this issue?
>
> **Answer**:
>
> We did not consider this perspective because the problem we aim to address is mainly defined by the scenarios in which users employ language models. We broadly categorized tasks into chatting, math, coding, and tool usage because these are the four most frequently encountered user scenarios. Moreover, switching the chunk-based attention in our method to an isolated attention mask would essentially revert back to standard SFT. Numerous studies have already demonstrated that language models are inherently multi-task learners. I believe that increasing the number of tasks will not affect the effectiveness of our strategy, as the simultaneous learning of math and coding tasks is already sufficiently challenging. Furthermore, our method enhances the model's ability to use code to solve math problems, which indirectly demonstrates the effectiveness of our approach.
>
> Our current packing length indeed exceeds the sequence length of the open-source SFT models themselves, so we have forcefully extended the context length of these models during training to 8192. We have already reported this in the experimental setup.
>
> **Weakness**:
>
> The description of the system prompt is somewhat confusing. In Section 2.4, the authors say that unique system prompts for each domain-specific task are set to ensure tailored guidance. However, in Table 1, they highlight that for all benchmark computations, they consistently use the system prompt "You are a helpful assistant." How can these conflict statements be understood? Does that mean the system prompts are different in training and testing?
>
> **Answer**:
> As illustrated in our Figure 3, during training, we deliberately divided the SFT training data into tasks across four domains. Some of this data inherently included a system prompt, and for these, we appended the corresponding description at the beginning of the existing system prompt. For data without a system prompt, we added one. Ultimately, all the training data was standardized into the chatml format.
>
> During evaluation as mentioned at ln306-310, we simply added the system prompt 'You are a helpful assistant.' to all validation datasets. This approach was chosen to genuinely test the practicality of this method, as in real-world applications where language models function as a service, it is impossible during a single inference to determine the exact task category of a user's query. Therefore, we made a simplistic abstraction by assuming that users will not modify the system prompt when invoking the language model. This requires our model to achieve the effect of task-specific system prompts under the default system prompt condition, which is why we only added the default system prompt during evaluation.
>
> **Weakness**:
> In section 2.2, the authors mention that training on combined data can lead to model bias toward certain tasks, enhancing performance in some while degrading it in others. This is related to the studies of catastrophic forgetting [1,2], which should be discussed in the related work or compared as baselines.
>
> **Answer**:
> Yes, the content you mentioned is indeed something we did not reference. We have added citations for these references in the new version of the paper.

---

> ### Author Response · Authors · 2024-11-26
> **Polite Reminder**
>
> Dear Reviewer ZeTr, I apologize for the interruption. Since another reviewer nEEX mentioned your concerns in his response, we kindly ask for your feedback on whether our rebuttal has addressed your concerns. Otherwise, this will hinder the communication between us and the other reviewer, which will seriously affect the review process of the paper.
>
> We look forward to your response.

---

> ### Author Response · Authors · 2024-11-27
> **Please check our response**
>
> Dear Reviewer ZeTr,
>
> Since the discussion period has been extended to December 3rd, but November 27th is the last day that authors may upload a revised PDF, we hope you can review our new version of the paper and the additional experimental content to see if they have addressed the weaknesses and questions you mentioned in your rebuttal.
>
> We look forward to your response.
>
> Thank you.

---

> > ### Comment · Reviewer_ZeTr · 2024-11-27
> > **Thank you for your responses!**
> >
> > Thanks for your detailed responses to my concerns. I think the paper is a good exploration of the empirical SFT process for many downstream tasks, and the experiments are solid. However, the contribution is still limited compared to a regular ICLR long paper and I will keep my score.

---

> > > ### Author Response · Authors · 2024-11-28
> > > **Thanks for your feedback!**
> > >
> > > Thank you for taking the time to help improve our paper.
> > >
> > > Your feedback has been truly helpful to us.
> > >
> > > Sincerely, thank you.

---

### Official Review · Reviewer_DmtX · 2024-11-04

**Soundness:** 4
**Presentation:** 4
**Contribution:** 3
**Rating:** 5
**Confidence:** 4

**Summary:**

This paper addresses a practical issue on multi-task supervised fine tuning (SFT). In order the model to be trained propoerly on all the tasks they propose packing the instructions belonging to different tasks in a balanced fashion. This results in superior performance on a number of tasks.

**Strengths:**

This is a clever and practical idea. The impact of this is clearly presented and I appreciate the comprehensive paper with many analyses.

**Weaknesses:**

However I feel like it may not constitute a full ICLR paper. Instead in the model system card, this could have been mentioned in a paragraph. For example Llama or Olmo model system cards have similar technical details like this.

**Questions:**

- Does the idea work for existing open SFT tuned models like Llama
- Does the idea apply to all model sizes and task categories?

---

> ### Author Response · Authors · 2024-11-22
> **Response to reviewer DmtX**
>
> Dear reviewer, thank you very much for your comments and professional advice. Based on your suggestions, we clarify some ambiguous parts of the paper and would like to provide the details as follows:
>
> **Weakness**:
>
> However I feel like it may not constitute a full ICLR paper. Instead in the model system card, this could have been mentioned in a paragraph. For example Llama or Olmo model system cards have similar technical details like this.
>
> **Answer**:
>
> It is true that some of our accompanying illustrations and experimental tables bear similarities to the model system cards of some excellent open-source language models on Hugging Face. This is because we both aim to provide a reliable alignment model, and in this process, the workload for validation is immense. A wide variety of datasets need to be tested to demonstrate the effectiveness of model alignment. This is actually a point we are quite pleased with in our work. We conducted extensive and detailed experiments to demonstrate that our method effectively improves performance across 10 validation sets, some of which include multiple tasks.
>
> If the concern is related to the formatting of our writing, we have expanded the main body of the paper from 9 pages to 10 pages in our newly submitted version, making our experimental conclusions and tables easier to read.
>
> If there are aspects of the method we proposed that are confusing, please specify your concerns in detail, and we will do our best to address your questions.
>
> **Question**:
>
> Does the idea work for existing open SFT tuned models like Llama
>
> **Answer**:
>
> In our paper, Table 6 already provides experimental results for Llama2-7B-chat and Llama3-8B-instruct, demonstrating that our method shows improvements on both models.
>
> **Question**:
>
> Does the idea apply to all model sizes and task categories?
>
> **Answer**:
>
> In Table 5 of our paper, we present the experimental results on 1.8B and 4B models, which demonstrate that our approach significantly improves performance on models of these sizes. Furthermore, the performance of small models trained with our Mixture of Instructions (MoI) approach is comparable to some excellent open-source small models, underscoring the effectiveness of our method.
> Regarding task categories, we initially defined the training dataset categories as code, math, chat, and tool usage, as these represent the domains where language models have substantial real-world applications (this is a non-academic definition). In our initial experiments, we tested the SFT performance on individual tasks, and ultimately, our MoI approach proved to be highly effective in leveraging data from all four tasks to comprehensively align the language model. Additionally, the experiments in Table 4 demonstrate that after simultaneously learning data from code, math, and tool usage tasks, our model gained the ability to use code to solve math problems.
>
> **Given that the questions and comments from the reviewer are quite brief, we are eager to receive more detailed feedback from you. We will do our best to address any questions or concerns you may have.**

---

> ### Author Response · Authors · 2024-11-26
> **Please check our response**
>
> Dear Reviewer DmtX,
>
> Since the discussion period has been extended to December 3rd, but November 27th is the last day that authors may upload a revised PDF, we hope you can review our new version of the paper and the additional experimental content to see if they have addressed the weaknesses and questions you mentioned in your rebuttal.
>
> We look forward to your response.
>
> Thank you.

---

> > ### Comment · Reviewer_DmtX · 2024-11-26
> >
> > Thanks, I am keeping my score.

---

> > > ### Author Response · Authors · 2024-11-27
> > > **Thanks for your feedback.**
> > >
> > > Thank you for taking the time to help improve our paper.
> > > This is good news for us!

---

### Official Review · Reviewer_Dhj1 · 2024-11-04

**Soundness:** 3
**Presentation:** 3
**Contribution:** 2
**Rating:** 6
**Confidence:** 4

**Summary:**

The paper introduces a technique known as Mixture-of-Instructions (MoI) for improving the alignment efficiency of large language models (LLMs) across multiple tasks. The technique uses instruction packing and diverse system prompts to enhance the model's performance. The authors applied this methodology to the Qwen-7B-chat model, resulting in the development of Qwen-SFT-MoI, which showed significant improvements in tasks like coding, mathematics, and tool use. The paper contributes by identifying and addressing conflicts between new and old knowledge during SFT, introducing the MoI method for joint multi-task training, and demonstrating the effectiveness of the MoI in enhancing SFT models.

**Strengths:**

1) The authors effectively identify a gap in existing alignment methodologies that typically focus on single task and propose a multi-task solution that is particularly relevant for AI-driven products.
2) The authors also provide a comprehensive evaluation of their method using seven benchmark datasets, demonstrating its effectiveness. The resulting model, Qwen-SFT-MoI, shows significant improvements in generative capabilities across multiple tasks.
3) The paper is well-written and clear in its presentation of the methodology and results. The work is significant as it offers a practical solution to a key challenge in the field of AI, namely the efficient alignment of LLMs for multi-task performance.

**Weaknesses:**

The primary concern revolves around the originality. The suggested MoI method incorporates several techniques, such as a distinct system prompt for each task, balanced sampling, and chunk-based attention. However, these techniques have already been introduced in previous research. For instance, the significance of the prompt for SFT has been highlighted by "Keeping llms aligned after fine-tuning: The crucial role of prompt templates, Lyu et al., 2024" and "Reducing the cost: Cross-prompt pre-finetuning for short answer scoring, Funayama et al., 2023". The technique of balanced sampling has been employed by "Sampling bias and class imbalance in maximum-likelihood logistic regression, Oommen et al., 2011", and the concept of chunk-based attention has been put forth by "Shifted chunk encoder for transformer based streaming end-to-end ASR, Wang et al., 2022" and "Statistically defined visual chunks engage object-based attention, Lengyel et al., 2021".

Considering these techniques are not innovative, the method seems more of an engineering blend if it fails to offer fresh perspectives or intriguing discoveries. Nevertheless, the positive outcomes and comprehensive experiments render the work robust and beneficial for the industrial sector.

**Questions:**

LN038: How did you come to the conclusion that the model generates less than ideal solutions due to conflicts in prompt-induced knowledge? Is there any research or experimental evidence to back this up?

LN040: Can you explain why modifying prompts can help in resolving knowledge conflicts?

LN085: What is the approach you use to identify conflicts between new and old knowledge during SFT? I couldn't find this information in section 2.

LN099: Are there any previous studies that highlight the significance of system prompts? If so, you should reference them and compare your method to theirs to demonstrate its superiority. If you're the first to do this, it's also worth mentioning.

LN470: The main focus of this study is multi-task learning, but I don't see a thorough discussion or comparison with existing methods.

---

> ### Author Response · Authors · 2024-11-25
> **Response to reviewer Dhj1**
>
> **Question**:
>
> LN099: Are there any previous studies that highlight the significance of system prompts? If so, you should reference them and compare your method to theirs to demonstrate its superiority. If you're the first to do this, it's also worth mentioning.
>
> **Answer**:
>
> [1] explores how incorporating character definitions in system prompts can assist language models in achieving multi-domain alignment while [2] ropose a novel persona-driven data synthesis methodology for SFT.  These studies demonstrate the role of character definitions in knowledge generation and task completion. [3] presents the first comprehensive cross-supervision alignment experiment in the role-play domain, revealing that the intrinsic capabilities of LLMs confine the knowledge within role-play.
>
> Based on these studies, we infer that the knowledge and expression of language models are related to role settings. Therefore, we attempt to modify roles to enable the language model to learn new knowledge during SFT。
>
> [1]Wang R, Mi F, Chen Y, et al. Role Prompting Guided Domain Adaptation with General Capability Preserve for Large Language Models[J]. arXiv preprint arXiv:2403.02756, 2024.
>
> [2]Ge T, Chan X, Wang X, et al. Scaling synthetic data creation with 1,000,000,000 personas[J]. arXiv preprint arXiv:2406.20094, 2024.
>
> [3]Lu K, Yu B, Zhou C, et al. Large language models are superpositions of all characters: Attaining arbitrary role-play via self-alignment[J]. arXiv preprint arXiv:2401.12474, 2024.
>
>
> **Question**:
>
> LN038: How did you come to the conclusion that the model generates less than ideal solutions due to conflicts in prompt-induced knowledge? Is there any research or experimental evidence to back this up?
>
> LN085: What is the approach you use to identify conflicts between new and old knowledge during SFT? I couldn't find this information in section 2.
>
> LN040: Can you explain why modifying prompts can help in resolving knowledge conflicts?
>
> **Answer**:
>
> In Figure 2, we demonstrate the issue of knowledge conflict by comparing how the attention of models trained with different system prompts changes on specific questions. In the original Chat model, attention highlights converge on words like "Boyer" and "track of frequency." For the model trained with the "You are a helpful assistant" prompt, attention highlights the word "iterate" on the same question. However, for the model trained with the "You are a programmer" system prompt, attention highlights fall on "Using the Boyer" and "keeping." Additionally, Figure 6 in the appendix documents the performance variance of the same Qwen-SFT-code model during inference with the "helpful assistant" and "programmer" prompts. This serves as further evidence of the model's differing performance under various roles and illustrates how our MoI method addresses this phenomenon. In the latest version of our paper, we have redrawn the significant attention score maps in Figure 2, which we consider noteworthy, to facilitate researchers in observing the performance changes of the model under different system prompts.
>
>
> **Question**:
>
> LN470: The main focus of this study is multi-task learning, but I don't see a thorough discussion or comparison with existing methods.
>
> **Answer**:
>
> [4] proposes concatenating multiple datasets during training and using focal loss to balance data bias among multiple tasks.
> [5] proved multi-task learning of LLM lead to conflicts, while sequential training results in catastrophic forgetting.
> [6] introduce Progressive Prompts, a continual learning method that learns a unique prompt for each new task while keeping the language model's parameters frozen. This approach prevents catastrophic forgetting and enables forward transfer by reusing information from prompts learned on previous tasks, serving as a good initialization for subsequent tasks.
>
> However, these studies have not addressed how to balance tasks and perform fine-tuning when language models are deployed as chat assistants that need to simultaneously meet four types of user needs: chatting, programming, mathematics, and tool invocation. In addition, we have even tested our approach in handling complex tasks where users request solving mathematical problems through code writing. This aspect has not been addressed in previous research on multitask learning with purely language models.
>
> [4]Liu B, Chen C, Gong Z, et al. Mftcoder: Boosting code llms with multitask fine-tuning[C]//Proceedings of the 30th ACM SIGKDD Conference on Knowledge Discovery and Data Mining. 2024: 5430-5441.
>
> [5]Dong G, Yuan H, Lu K, et al. How abilities in large language models are affected by supervised fine-tuning data composition[J]. arXiv preprint arXiv:2310.05492, 2023.
>
> [6]Razdaibiedina A, Mao Y, Hou R, et al. Progressive prompts: Continual learning for language models[J]. arXiv preprint arXiv:2301.12314, 2023.
>
> **We have added these references to the related work section in the new version of the paper.**

---

> > ### Comment · Reviewer_Dhj1 · 2024-11-27
> > **Thanks for clarification**
> >
> > I appreciate the authors for their comprehensive explanation. I would prefer to maintain my existing positive rating.

---

> > > ### Author Response · Authors · 2024-11-28
> > > **Thanks for your feedback!**
> > >
> > > Thank you for taking the time to help improve our paper.
> > >
> > > Your feedback has been truly helpful to us.
> > >
> > > Sincerely, thank you.

---

### Author Response · Authors · 2024-11-25
**Rebuttal Revision Version Paper Modifications**

We thank the reviewers for their time and effort on this work. Based on their feedback, we have made the following revisions to the paper:

1.To address reviewer DmtX's concern that our paper resembled a model system card, **we reformatted the experimental and analysis sections, expanding the overall length from 9 to 10 pages**. In the revised paper, each discussion point in the analysis section is accompanied by the relevant experimental results on the same page, reducing the need for frequent page-turning while reading.

2.We have added more references on multi-task learning to the original related work section.

3.We included a new related work section titled "SFT in the alignment of LLM," providing references on catastrophic forgetting during SFT and the impact of system prompts on language models during the SFT process.

4.We have redrawn Figure 2, adding arrows to indicate key attention responses.

5.We included experimental results related to RAG and Qwen72B in the appendix, addressing the concerns of reviewer bxFP.

6.We added supplementary experiments in the appendix to address reviewer nEEX's concerns about "Why is the attention of MoI better?"

We appreciate all the valuable feedback provided by the reviewers at this stage. As the rebuttal deadline is approaching, we have responded to all the reviewers. If there are still any concerns, please let us know as soon as possible.

---

### Meta-Review · Area_Chair_FYX8 · 2024-12-22

**Metareview:**

This paper proposes an approach, Mixture-of-Instructions (MoI), that aims to align a model with several tasks simultaneously. The approach is based on instruction packing with diverse system prompts to represent and improve multiple downstream tasks during supervised fine-tuning. Authors test the approach on a new benchmark they compiled from existing benchmarks, and create the MoI version of the Qwen-7B-chat model. Reviewers highlight the contributions as the ease of applying the approach and comprehensive experimental results. The weaknesses include the limited novelty based on the incremental nature of the approach and limited experimentation, for example, the analysis on the single model.

**Additional Comments On Reviewer Discussion:**

Rebuttal period included good discussion between the authors and the reviewers. Authors responded positively to the suggestions and included additional results with new models. However, there were still more questions unanswered for the reviewers at the end of the discussions.

---

### Decision · Program_Chairs · 2025-01-22

Reject